# Multidecadal increases in global tropospheric ozone derived from ozonesonde and surface site observations: Can models reproduce ozone trends?

Amy Christiansen[1][†], Loretta J. Mickley[2], Junhua Liu[3,4], Luke D. Oman[4], Lu Hu[1]

[1]Department of Chemistry and Biochemistry, University of Montana, Missoula, MT, 59812, USA
[†]Current address: Division of Energy, Matter & Systems, University of Missouri – Kansas City, Kansas City, MO 64110
[2]John A. Paulson School of Engineering and Applied Sciences, Harvard University, Cambridge, MA, 02138, USA
[3]Morgan State University, Baltimore, Maryland, 21251, USA
[4]Atmospheric Chemistry and Dynamics Laboratory, NASA Goddard Space Flight Center, Greenbelt, MD, 20771, USA

*Correspondence to*: Lu Hu (lu.hu@mso.umt.edu); Amy Christiansen (achristiansen@umkc.edu)

**Abstract.** Despite decades of effort, the drivers of global long-term trends in tropospheric ozone are not well understood, impacting estimates of ozone radiative forcing and the global ozone budget. We analyze tropospheric ozone trends since 1980 using ozonesondes and remote surface measurements around the globe and investigate the ability of two atmospheric chemical transport models, GEOS-Chem and MERRA2-GMI, to reproduce these trends. Global tropospheric ozone trends measured at 25 ozonesonde sites from 1990-2017 (9 sites since 1980s) show increasing trends averaging $1.8 \pm 1.3$ ppb decade$^{-1}$ across sites in the free troposphere (800-400 hPa). Relative trends in sondes are more pronounced closer to the surface (3.5% decade$^{-1}$ above 700 hPa, 4.3% decade$^{-1}$ below 700 hPa on average), suggesting the importance of surface emissions (anthropogenic, soil NO$_x$, impacts on biogenic VOCs from land use changes, etc.) in observed changes. While most surface sites (148 of 238) in the United States and Europe exhibit decreases in high daytime ozone values due to regulatory efforts, 73% of global sites outside those regions (24 of 33 sites) show increases from 1990-2014 that average $1.4 \pm 0.9$ ppb decade$^{-1}$. In all regions, increasing ozone trends both at the surface and aloft are at least partially attributable to increases in 5$^{th}$ percentile ozone, which average $1.8 \pm 1.3$ ppb decade$^{-1}$ and reflect the global increase of baseline ozone in rural areas. Observed ozone percentile distributions at the surface have shifted notably across the globe: all regions show increases in low tails (i.e., below 25$^{th}$ percentile), North America and Europe show decreases in high tails (above 75$^{th}$ percentile), and the Southern Hemisphere and Japan show increases across the entire distribution. Three model simulations comprising different emissions inventories, chemical schemes, and resolutions, sampled at the same locations and times of observations, are not able to replicate long-term ozone trends either at the surface or free troposphere, often underestimating trends. We find that ~75% of the average ozone trend from 800-400 hPa across the 25 ozonesonde sites is captured by MERRA2-GMI and <20% is captured by GEOS-Chem. MERRA2-GMI performs better than GEOS-Chem in the northern mid-latitude free troposphere, reproducing nearly half of increasing trends since 1990 and capturing stratosphere-troposphere exchange (STE) determined via a stratospheric ozone tracer. While all models tend to capture the direction of shifts in the ozone distribution and typically capture changes in high and low tails, they tend to underestimate the magnitude of the shift in medians. However, each model shows an 8-12%

(or 23-32 Tg) increase in total tropospheric ozone burden from 1980 to 2017. Sensitivity simulations using GEOS-Chem and the stratospheric ozone tracer in MERRA2-GMI suggest that in the northern mid- and high latitudes, dynamics such as STE are most important for reproducing ozone trends in models in the middle and upper troposphere, while emissions are more important closer to the surface. Our model evaluation for the last 4 decades reveals that the recent version of the GEOS-Chem model underpredicts free tropospheric ozone across this long time period, particularly in winter and spring over mid-to high latitudes. Such widespread model underestimation of tropospheric ozone highlights the need for better understanding of the processes that transport ozone and promote its production.

## 1 Introduction

Tropospheric ozone is an air pollutant detrimental to human and vegetative health, with increased levels at the surface linked to morbidity, premature mortality (Monks et al., 2015; Bell et al., 2006), and damage to plant structures and productivity (Ainsworth et al., 2012; Mills et al., 2018). In the upper troposphere, ozone interacts with both incoming solar radiation and outgoing longwave radiation, thus acting as a strong greenhouse gas (Monks et al., 2015; Forster et al., 2007). Its spatial and temporal heterogeneity make it a powerful yet highly uncertain regional climate forcer (Naik et al., 2005; Worden et al., 2008). Ozone plays an important role in tropospheric oxidation capacity through its influence on radical cycles and lifetimes of other atmospheric pollutants (Stone et al., 2012), including secondary aerosols (Karset et al., 2018). At the same time, ozone production is dependent on those radical cycles. Tropospheric ozone is produced via the photooxidation of methane ($CH_4$), volatile organic compounds (VOCs), and carbon monoxide (CO) in the presence of nitrogen oxides ($NO_x$). Ozone concentrations are also dependent on temperature, water vapor, and large-scale dynamics (Griffiths et al., 2020; Pusede et al., 2015; Steiner et al., 2006; Lin et al., 2020, 2014). The average lifetime of ozone in the troposphere is about 3 weeks, allowing it to be transported laterally (Lin et al., 2017) and from the stratosphere to the troposphere through stratosphere-troposphere exchange (STE) (Griffiths et al., 2020; Williams et al., 2019; Gettelman et al., 1997; Sullivan et al., 2015). Despite decades of effort, the drivers of global long-term trends in ozone are not well understood. We seek in this work to quantify observed global ozone trends since 1980 using ozonesondes and surface measurements, and we investigate the ability of two atmospheric chemical transport models, GEOS-Chem and MERRA2-GMI, to reproduce these trends.

Observations from ground stations, ozonesondes, and satellites have indicated that overall global tropospheric ozone has been increasing in recent decades throughout the troposphere (Ziemke et al., 2019; Cooper et al., 2020; Gaudel et al., 2020; Lu et al., 2019; Archibald et al., 2020; Cooper et al., 2014). A subset of models used in the Chemistry Climate Model Initiative (CCMI) intercomparison simulations estimated an approximate increase in tropospheric ozone burden of 50 Tg from 1960-2010 (Morgenstern et al., 2017), and a simulation with the chemistry-climate model CAM-chem suggested an increase of 28 Tg from 1980-2010 (Zhang et al., 2016). Confirmation of model results using in situ observations is challenging due to sparse

65 measurements, but satellite measurements improve on these spatial limitations. From 1997-2014, measurements from satellite ensembles estimated changes in tropospheric ozone burden of 15 Tg between 60°S-60°N (Griffiths et al., 2021). Both modeled and observed increases in the global burden of tropospheric ozone have been attributed to multiple factors, including an equatorward redistribution of emissions, where meteorological factors such as ultraviolet radiation and water vapor allow for increased photochemical production in the tropics and subtropics (Zhang et al., 2021, 2016).

70

Free tropospheric (FT) ozone changes are highly regional and are impacted by emissions and transport. Aircraft measurements from 1995-2015 suggest FT ozone has increased strongly over Southeast Asia (5.6 ppb decade$^{-1}$; 14% decade$^{-1}$) (Gaudel et al., 2020), which is largely attributed to emissions increases. Ziemke et al. (2019) found that ozone increased over East Asia by 1 DU decade$^{-1}$ from 1979-2005 (~25% decade$^{-1}$) via satellite measurements, consistent with Ding et al. (2008), who found that 
75 ozone increased over Beijing by 20% decade$^{-1}$ from 1995-2005 using aircraft measurements. Increases over Asia have occurred most rapidly starting in the mid-2000s (~6% yr$^{-1}$; ~60% decade$^{-1}$) (Oetjen et al., 2016; Ziemke et al., 2019). Differences in trends between these studies can be attributed to differences in geographical areas (e.g., Beijing vs. Southeast Asia) as well as date ranges. Transport of ozone from Asia impacts ozone trends in other regions, and this is estimated to have offset 43% of the expected reduction in FT ozone over the western United States from 2005-2013 (Verstraeten et al., 2015). Aircraft 
80 measurements have also noted weak ozone increases in the northeastern US and German FT of <7% decade$^{-1}$ (<5 ppb decade$^{-1}$) (Gaudel et al., 2020). Over the Southern Hemisphere, ozonesonde measurements show an increase in ozone from 1990-2015, which is linked to both increasing precursor emissions and large-scale dynamics such as STE (Lu et al., 2019; Zeng et al., 2017). Ozone measurements from the Southern Hemisphere Additional Ozonesondes (SHADOZ) network show increasing FT ozone in some parts of the tropics, with tropical South American and Asian sites showing annual average increases of 5% 
85 decade$^{-1}$ from 1998-2019 (Thompson et al., 2021). However, large regions of the tropics do not show annual increases, with increasing ozone limited to certain seasons at most stations (e.g., Nairobi, Kenya, FT ozone increases 5-10% decade$^{-1}$ during February-April but does not increase on an annual basis) (Thompson et al., 2021).

STE has also been shown in both observations and models to have a substantial impact on tropospheric ozone trends and 
90 interannual variability, with stratospheric intrusion events influencing decadal trends across North America, Europe, the Southern Pacific, and the southern Indian Ocean (Williams et al., 2019; Liu et al., 2020). For example, models suggest that 25-30% of increases in surface ozone between 1980 and 2010 were attributable to STE in multiple regions (Williams et al., 2019) and >10% of interannual variability in surface ozone could be explained by stratospheric ozone in winter and spring in North America (Liu et al., 2020).

95

Surface ozone trends are largely driven by local emissions, and the direction and magnitude of trends relies on local changes and regulations. The largest increases in surface ozone over the past few decades have occurred over Asia (up to 6 ppb decade$^{-1}$), where a tripling of NO$_x$ since 1990 has led to large increases in surface ozone over the region (Ziemke et al., 2019; Lin et

al., 2017). Over China, despite substantial decreases in $NO_x$ emissions in recent years, maximum 8-hour average ozone concentrations have increased by 1.9 ppb $yr^{-1}$ as a result of decreased concentrations of $PM_{2.5}$, which scavenges radicals needed for ozone formation (Li et al., 2020). Over the western US and Europe, ozone increases over Asia since the 1990s have increased low-percentile surface ozone levels due to hemispheric transport (Cooper et al., 2012; Yan et al., 2018a; Lawrence and Lelieveld, 2010). However, peak surface ozone values have decreased over these regions due to regulations, as these values are more sensitive to local emissions than transport (Fiore et al., 2014; Lin et al., 2017; Yan et al., 2018a).

Despite being the subject of intensive study, many questions regarding the global tropospheric ozone trend remain. Much of the evidence around tropospheric ozone changes has come from the analysis of surface ozone trends, especially over the United States and Europe (Yan et al., 2018a, b; Lefohn et al., 2010; Yan et al., 2018b; Simon et al., 2015). Changes over these regions since the 1990s are often characterized by shifts in the magnitude of the seasonal cycle (Bowman et al., 2022) and decreasing peak and summertime ozone values, in contrast to the increasing annual mean ozone driven by increasing low-percentile (e.g., 5th and 10th percentile) ozone. However, changes occurring at the surface may differ from changes above the boundary layer due to the increased importance of transport processes over emissions in the FT. Trends throughout the troposphere can be investigated via satellites measuring total ozone throughout the entire atmospheric column after accounting for the stratospheric contribution. However, they do not allow for analyses of trends at different pressure levels and are subject to uncertainties stemming from approaches to remove stratospheric ozone from total column measurements (Liu et al., 2010; Ziemke et al., 2019, 2011). Aircraft data from the IAGOS (In-Service Aircraft for a Global Observing System) have been used for ozone trends at different pressure levels (Petzold et al., 2015). While useful, these vertical profiles are taken near airports, and data are only available starting in the mid-1990s. Ozonesondes represent an underutilized dataset that allows for the analysis of ozone trends at multiple pressure levels throughout the troposphere and beyond (Thompson, 2003; Thompson et al., 2004, 2007, 2011; von der Gathen et al., 1995; WMO, 1998). Ozonesondes improve upon the vertical resolution limitations of satellites, and several sites around the globe have measured ozone since the 1980s or earlier. While it is not reasonable to extrapolate sparsely located ozonesonde measurements to changes occurring on all parts of the globe, ozonesondes are essential to understanding trends at distinct vertical levels since these are the only technique capable of measuring ozone concentrations from near the surface and into the stratosphere while maintaining high accuracy and vertical resolution (Van Malderen et al., 2021).

Previous literature focusing on ozonesonde trends has often focused on specific regions or individual sonde launch locations. In many applications, ozonesonde information is used to validate or assess satellite retrievals rather than as a primary source to investigate trends (Boynard et al., 2018; Shi et al., 2017; Hulswar et al., 2020; Huang et al., 2017; Bak et al., 2019). To date, the most extensive look at ozonesonde trends over Europe is provided by Logan et al. (2012), where trends up to 2011 were evaluated. In that analysis, the authors found that ozone increased over Europe during the 1990s and then decreased during the 2000s. Over the Southern Hemisphere, trends in ozone using ozonesondes have been analyzed at several locations from 1990-

2015, focusing on increases in austral autumn (Lu et al., 2019). At Arctic sites, ozone at all pressure levels increased from the late 1980s until 2005, then decreased (Christiansen et al., 2017). Trends from ozonesondes over Canada show mixed results, where one analysis found ozone increases from 2005-2014 (Christiansen et al., 2017), and another found no significant trend from 1966-2013 (Tarasick et al., 2016). These differences are partially attributable to a difference in analysis timeframes. Ozonesonde analyses in East Asia have found strong increases since the 2000s (Lin et al., 2017; Zhu et al., 2017). In this work, we combine long-term continuous ozonesonde measurements from global sites across a consistent timeframe to allow for a better perspective on long-term (30+ years) global tropospheric ozone changes occurring at distinct vertical levels throughout the troposphere.

Understanding the long-term trends in tropospheric ozone concentrations is critical for accurately estimating ozone radiative forcing, policy-relevant ozone background (ozone concentrations in the absence of anthropogenic emissions), and global tropospheric hydroxyl radical concentrations. Even for recent decades, large uncertainties exist in model estimates of ozone burden change and radiative forcing. The radiative forcing due to the 1850-present day change in tropospheric ozone has been estimated to be +0.16 to +0.49 W m$^{-2}$ (Checa-Garcia et al., 2018), a range corroborated by a recent multi-model intercomparison (+0.29 to +0.53 W m$^{-2}$) (Skeie et al., 2020). The most recent multi-model study investigating short-term ozone changes from 1990-2015 yielded a mean ozone forcing of +0.06 W m$^{-2}$ (Myhre et al., 2017), about 50% greater than a previous estimate over the same timeframe (Myhre et al., 2013) and a more recent estimate from 2010-2018 (Skeie et al., 2020). The greater value in Myhre et al. (2017) has been attributed to the greater increase in $NO_x$ emissions in that estimate. Parrish et al. (2014) and Staehelin et al. (2017) showed that four state-of-the-science chemistry-climate models overestimate the absolute ozone mixing ratio by 5-17 ppb at mid-latitude background sites and capture only about half of the observed ozone increase over the last five decades, casting doubt on estimates of even the short-term radiative effect of changing ozone. Representativeness of ozone measurements, especially those made prior to satellite information, is one of the leading challenges in model reproduction of ozone trends and understanding of ozone radiative forcing (Tarasick et al., 2019). Most estimates of ozone radiative forcing are calculated relative to the pre-industrial period (Skeie et al., 2020; Stevenson et al., 2013), and the small amount of reliable measurements prior to the 20$^{th}$ century (Tarasick et al., 2019) provides challenges to constraining both long and short-term radiative estimates. Even short-term changes in ozone can be difficult to reproduce, as a recent chemical transport model simulation of global ozone trends over the past ~20 years show a consistent underestimate of observed ozone trends (Wang et al., 2022).

As analytical techniques for ozone measurements today are more robust than in the 19$^{th}$ and early 20$^{th}$ centuries, the inability of models to capture recent decadal trends of tropospheric ozone is concerning. A range of common model issues or observational limitations have been suggested as the causes of these discrepancies, as summarized by the Tropospheric Ozone Assessment Report (TOAR) (Tarasick et al., 2019; Young et al., 2018). These include uncertainties in early ozone measurements stemming from analysis techniques, temporal and spatial mismatches between observations and model output,

the use of "freely running" chemistry-climate models which cannot represent actual meteorological conditions, and errors in model emission inventories (Logan et al., 2012; Cooper et al., 2014; Lin et al., 2014; Strode et al., 2015; Hassler et al., 2016; Lin et al., 2017; Staehelin et al., 2017; Koumoutsaris and Bey, 2012; Barnes et al., 2016). Recent model advances targeting anthropogenic emissions, lightning emissions, halogen chemistry, isoprene chemistry, and assimilation of observed meteorological fields have overall led to more active ozone chemistry in models (Hu et al., 2017). Such increasingly active tropospheric chemistry in models affects ozone sensitivity to emission perturbations, impacting simulated ozone changes over time. For example, the implementation of halogen chemistry in GEOS-Chem reduced ozone radiative forcing estimates since the preindustrial era by more than 20% (Sherwen et al., 2017). Further, emissions estimates of important ozone precursor species are subject to many uncertainties, including the magnitude of emissions activities and scaling factors applied at local and regional scales. Previous analyses have found that models overestimate $NO_x$ in the United States and India (McDonald et al., 2013, 2018; Anderson et al., 2014; Ghude et al., 2013) but underestimate this species in Europe (Terrenoire et al., 2015; Mar et al., 2016). Assessments of emissions inventories are difficult in regions that do not have reliable ground-based measurements such as rapidly developing areas in Latin America and Africa (Hassler et al., 2016). The inability of a wide variety of models to capture ozone concentrations and trends on multiple time scales indicates large uncertainties in our understanding of tropospheric ozone and its implications for radiative forcing and air quality regulations.

In this work, we explore long-term trends in ozone concentrations from 1980-2017 at multiple vertical levels throughout the troposphere using global individual ozonesonde stations and surface ozone monitoring sites. We also assess the ability of three global simulations from two chemical transport models (CTMs) comprising different emissions inventories, chemical schemes, and resolutions to reproduce long-term trends at the surface and aloft from 1980-2017, with implications for understanding ozone radiative forcing, tropospheric ozone budget, and policy-relevant background ozone. These models represent the state of the science, including the most updated emissions inventories, recent updates to chemical mechanisms, and assimilated meteorological fields. To obtain the best comparison of ozone concentrations and trends, we sample each model at ozonesonde launch times and locations, a step not often taken in ozone model-measurement comparisons. We also attempt to identify potential reasons for model-measurement discrepancies.

## 2 Methods

### 2.1 Observational Datasets

Ozonesonde vertical profile measurements from 1980-2017 were downloaded from the World Ozone and Ultraviolet Data Center (WOUDC) (https://woudc.org/data/explore.php), the National Oceanic and Atmospheric Administration (NOAA) (ftp://ftp.cmdl.noaa.gov/ozww/Ozonesonde/), and the Harmonization and Evaluation of Ground-based Instruments for Free Tropospheric Ozone Measurements (HEGIFTOM) working group of the Tropospheric Ozone Assessment Report, Phase II (TOAR-II) (https://hegiftom.meteo.be/datasets/ozonesondes). The global ozonesonde community is currently reprocessing

and homogenizing data to account for changes in ozonesonde preparation and procedures, with the goal to reduce measurement biases associated with these changes (Tarasick et al., 2016; Van Malderen et al., 2016; Witte et al., 2018; Sterling et al., 2018; Ancellet et al., 2022). Where possible (12 of 25 sites), homogenized ozone profiles were used to ensure the most accurate ozone trends. Table 1 describes the ozonesonde profile information, dates, and whether the data is homogenized. While Payerne (Europe) has homogenized data, we use the original data since the site has only been homogenized since 2002. For data that is not homogenized, we ensure that it does not contain step changes (Figs. S1 & S2). Updated tropical ozonesonde information is available from the Southern Hemisphere Additional OZonesondes (SHADOZ) (https://tropo.gsfc.nasa.gov/shadoz/), but we did not include these data in this analysis because they did not meet our data requirements described below, typically due to not having enough profiles per month consistently throughout our timeframe.

Most ozonesonde data were measured by electrochemical concentration cell (ECC) sensors, widely regarded as the most accurate sensor type (Tarasick et al., 2021). Four sites (Payerne, Uccle, Legionowo, Lindenberg) in Europe switched from using Brewer-Mast (BM) sensors to ECC sensors partway through their data records, and data from both sensors were used since previous analyses showed good agreement between measurements (De Backer et al., 1998; Stübi et al., 2008). Only Hohenpeissenberg (Europe) used the BM sensor throughout the time period. Naha (Japan), Tsukuba (Japan), Sapporo (Japan), and Syowa (Antarctica) both used carbon iodine (CI) sensors prior to 2010 and ECC sensors after, and this switch could impact overall long-term trends (Tanimoto et al., 2015). Typical uncertainties for CI sensors range from 5-10%, while they are 3-5% for ECC sensors (Tanimoto et al., 2015). This could lead to substantial differences in calculated trends, and we discuss trends from these sites in the context of regional trends using sites with more reliable data (e.g., only one sensor type or homogenized data). We note that trends at these sites should be treated with caution. A recent study showed a drop in total column and stratospheric ozone measured by ECC instruments compared to satellite observations in the latter parts of their records for reasons still under investigation (Stauffer et al., 2020, 2022). We find that 5 of our 25 sites were impacted by these ozone measurement drops, although these drop-offs were typically limited to pressures above ~50 hPa, so our results should not be affected. Out of an abundance of caution, at these impacted sites, we used only data from before the unexplained sharp drop-off in ozone concentrations, as data before these drops is still considered highly reliable (Stauffer et al., 2020, 2022), and this resulted in the removal of up to one year of data at each affected site.

Ozonesonde profiles were reduced to match the 47-layer GEOS-Chem reduced pressure levels by aggregating all observed ozone values between model-defined pressure edges. The following criteria for ozonesonde sites were used in this analysis from 1990-2017. Locations were selected based on data completion criteria adapted from Lu et al. (2019): 1) at least 3 observations per month, 2) at least 2 monthly observations per season, 3) at least 8 monthly observations per year, and 4) at least 16 years of data. These data requirements were met by 25 ozonesonde locations throughout the globe for the 1990-2017 time periods (Fig. 1). Nine of the selected sites have data extending back to the 1980s, and these trends are discussed where appropriate, although the main focus of this work is on trends after 1990.

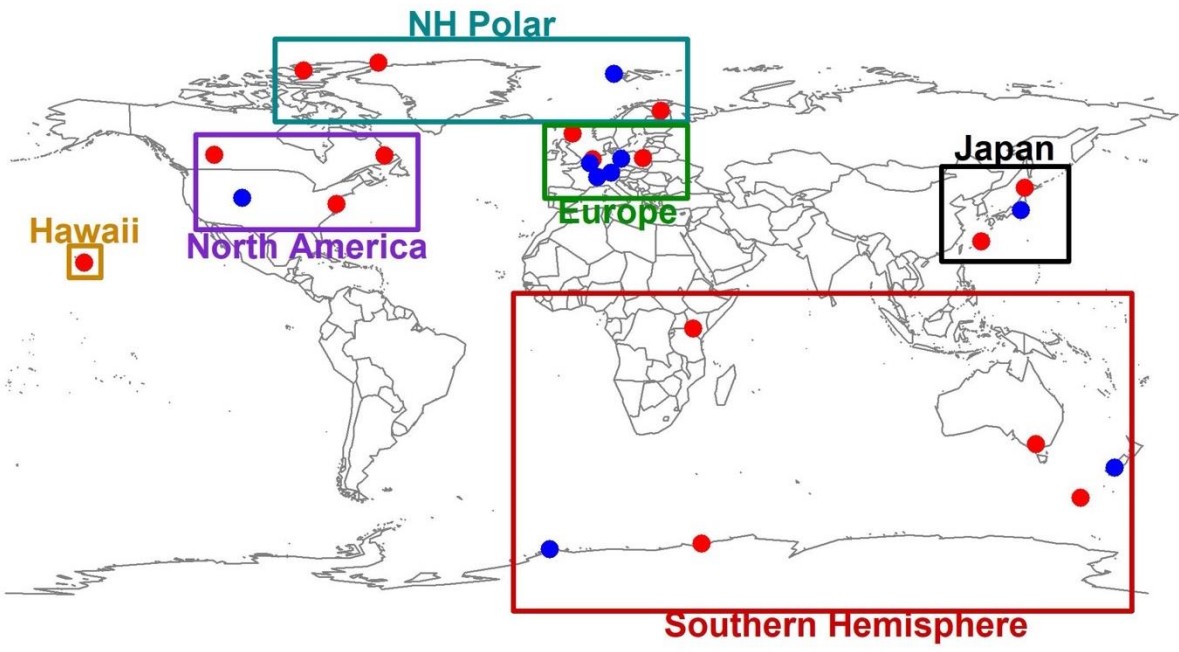

Figure 1. Map showing ozonesonde locations. Locations with data spanning 1990-2017 are shown in red, and locations with data extending to the 1980s are shown in blue. The boxes represent the regions into which all ozonesondes are grouped.

Table 1. Summary of all ozonesonde launch locations, dates, sensor types, data source, and region. Also included is whether each site has been homogenized.

| Sonde Launch Location | Dates | Sensor Type | Homogenized? | Data Source | Region |
|---|---|---|---|---|---|
| Alert | 1990-2016 | ECC | Y | HEGIFTOM | NH Polar |
| Boulder | 1980-2016 | ECC | Y | NOAA | North America |
| Broadmeadows | 1999-2016 | ECC | N | WOUDC | Southern Hemisphere |
| De Bilt | 1993-2015 | ECC | Y | HEGIFTOM | Europe |
| Edmonton | 1980-2016 | ECC | Y | HEGIFTOM | North America |
| Eureka | 1993-2016 | ECC | Y | HEGIFTOM | NH Polar |

| | | | | | |
|---|---|---|---|---|---|
| Goose Bay | 1980-2016 | ECC | Y | HEGIFTOM | North America |
| Hilo | 1985-2015 | ECC | Y | SHADOZ | Hawaii |
| Hohenpeissenberg | 1980-2017 | BM | Y | HEGIFTOM | Europe |
| Lauder | 1986-2016 | ECC | Y | HEGIFTOM | Southern Hemisphere |
| Legionowo | 1980-2015 | BM, ECC since 1993 | N | WOUDC | Europe |
| Lerwick | 1994-2016 | ECC | N | WOUDC | Europe |
| Lindenberg | 1980-2013 | BM, ECC since 1992 | N | WOUDC | Europe |
| Macquarie Island | 1994-2017 | ECC | N | WOUDC | Southern Hemisphere |
| Naha | 1991-2016 | CI, ECC since 2008 | N | WOUDC | Japan |
| Nairobi | 1998-2016 | ECC | Y | SHADOZ | Southern Hemisphere |
| Neumayer | 1992-2014 | ECC | N | WOUDC | Southern Hemisphere |
| Ny Aalesund | 1990-2012 | ECC | N | WOUDC | NH Polar |
| Payerne | 1980-2016 | BM, ECC after 2002 | Y* | HEGIFTOM | Europe |
| Sapporo | 1993-2016 | CI, ECC since 2009 | N | WOUDC | Japan |
| Sodankyla | 1989-2006 | ECC | N | WOUDC | NH Polar |
| Syowa | 1982-2017 | CI, ECC since 2010 | N | WOUDC | Southern Hemisphere |
| Tateno | 1980-2016 | CI, ECC since 2009 | N | WOUDC | Japan |

| Uccle | 1980-2015 | BM, ECC since 1997 | Y | HEGIFTOM | Europe |
|---|---|---|---|---|---|
| Wallops Island | 1995-2016 | ECC | Y | HEGIFTOM | North America |

*Note that Payerne has been homogenized only since 2002, a timeframe too short for this analysis, so we use the original data that spans the full timeframe.

Surface daytime baseline ozone data from 1990-2014 were obtained from the TOAR Surface Ozone Database (Schultz et al., 2017), which has been compiled and processed by the TOAR Database team and made public via
https://doi.org/10.1594/PANGAEA.876108. Each site in this database has at least 70% of all hourly ozone measurements available for each year provided as monthly aggregates. Similar to the ozonesondes, sites used in this analysis were constrained by the following criteria: 1) at least 2 monthly observations per season, 2) at least 8 monthly observations per year, and 3) at least 15 years of data throughout the timeframe. TOAR site locations are shown in Fig. 2 below. All sites are in background locations, which is defined by individual data providers to the TOAR database with no formal unifying definition (Schultz et
al., 2017). All sites were also classified as "rural," which is defined as: 1) $NO_2$ column $\leq 8x10^{15}$ molecules cm$^{-2}$ as measured by the Ozone Monitoring Instrument (OMI), 2) an averaged nighttime light intensity index of $\leq 25$ within a 5 km radius of the site, and 3) a maximum population density of $\leq 3000$ people km$^{-2}$ within a 5 km radius of the site (Schultz et al., 2017). Of the 271 surface site locations meeting these requirements, 52 site locations are in the United States and 173 are in Europe, biasing trend information to these areas (Fig. 2). However, there are 33 background sites in other regions spanning the globe that give
insight to changes in surface ozone beyond the northern mid-latitudes.

Included in these sites are 8 high elevation sites (>2800 m), which are discussed separately from the other surface sites and are marked with blue dots in Fig. 2. These sites include 5 mountaintop sites, which have been studied extensively to determine if ozone trends at these sites are dominated by FT air (Logan et al., 2012; Parrish et al., 2014; Cooper et al., 2020), generally by
260 using nighttime ozone values to avoid influence from local air masses. During the day, mountaintops often experience updrafts of polluted air from lower altitudes. While these sites have traditionally been used as another method for identifying lower FT ozone trends (Logan et al., 2012; Parrish et al., 2014), a recent analysis of 3 European mountaintop sites (Jungfraujoch, Sonnblick, and Zugspitze) found they were influenced by boundary layer air and were thus more representative of the lower troposphere (Cooper et al., 2020). Other mountaintop sites (Mauna Loa and Mt. Waliguan) have been found to be
representative of FT air when the data is filtered appropriately to exclude air masses influenced by the boundary layer (Cooper et al., 2020; Lin et al., 2014; Xu et al., 2018). Here, we did not seek to reiterate trends since the 1990s reported in previous studies, but rather used these high elevation and mountaintop sites representative of regional or FT air to corroborate observed ozonesonde trends. Six of the sites (Centennial, Gothic, South Pole, Zugspitze, Jungfraujoch, and Sonnblick) were used as a

point of comparison for lower tropospheric (>700 hPa) ozonesonde trends, and two sites (Mauna Loa and Mt. Waliguan) were
used for FT ozonesonde trends (700-400 hPa). Trends from each site were reported using ozone measurements from various
times during the 24-hour diurnal cycle to capture regional or FT trends, and the times used are specified in Sect. 3.4.

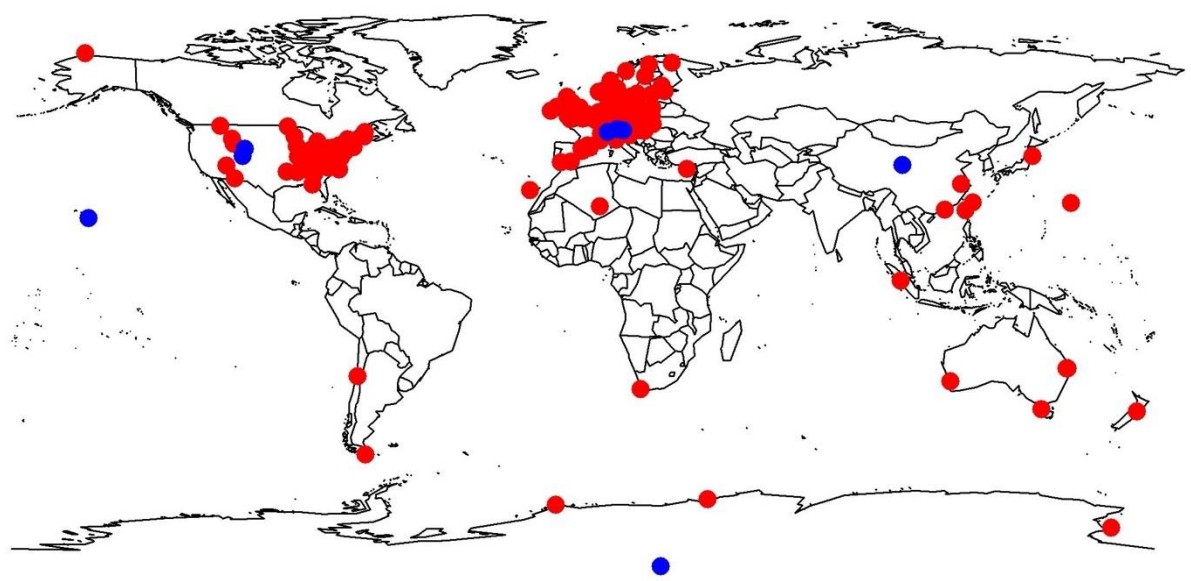

**Figure 2. Surface site locations of baseline ozone monitors with data spanning 1990-2014, compiled and processed by the TOAR**
**(Tropospheric Ozone Assessment Report) Database team. High elevation sites (>2800 m a.s.l.) that represent the lower troposphere**
**or FT are shown in blue.**

Ozonesonde and TOAR surface data were analyzed using R statistical software (R Core Team, 2013). In this work, we reported
trends as ppb decade$^{-1}$ and considered them significant if p<0.1. Trends were calculated using deseasonalized data and quantile
regression due to the intermittent nature of the ozonesonde launches (Gaudel et al., 2020; Koenker and Bassett, 1978).
Deseasonalization reduces the impact of autocorrelation. At each pressure level and site, we constructed a mean seasonal cycle
for each site's timeframe. This seasonal cycle was then used to deseasonalize individual observations on each pressure level.
Quantile regression is an expansion of linear regression which predicts trends for a distribution rather than using conditional
means. An advantage of quantile regression for our dataset is that it does not require the aggregation of sparse data to monthly
means. As most ozonesonde locations launch only 3-4 times each month, monthly mean values may not be statistically
meaningful. Quantile regression is also robust for datasets containing outliers and intermittent missing values, making it
appropriate for our ozonesonde dataset. Quantile regression has the added benefit of predicting trends for various percentiles
of the distribution, allowing for the examination of extreme trends (e.g., 5$^{th}$ percentile). Linear trends were calculated using all
profiles in the timeframe.

**2.2 Model Configurations**

To evaluate model ability to reproduce long-term ozone trends, we analyzed a variety of model configurations comprising different emissions inventories, chemical schemes, and resolutions. We used two simulations of GEOS-Chem v12.9.3 (GC) and a replay simulation of the National Aeronautics and Space Administration Goddard Earth Observing System (NASA
GEOS) model coupled to the Global Model Initiative (GMI) chemical mechanism and meteorological information from MERRA-2 reanalysis data, hereafter referred to as MERRA2-GMI. We also used a shorter simulation from an earlier version of GEOS-Chem (v10-01) that spans 1980-2010 as a point of comparison. The details for each of these simulations are described below and in Table 2.

**Table 2. Description of three simulations with GEOS-Chem version 12 (two simulations at different resolution; GC 4x5 and GC 2x2.5) and MERRA2-GMI.**

| Model | GEOS-Chem version 12 (GC 4x5 and GC 2x2.5) | NASA MERRA2-GMI[a] (MERRA2-GMI) |
|---|---|---|
| **Horizontal resolution (latitude x longitude)** | 4°x5° & 2°x2.5° | 0.5°x0.625° |
| **Chemistry** | v12.9.3[b] | GMI[c] |
| **Meteorology** | Modern-Era Retrospective analysis for Research and Applications version 2 (MERRA-2) | Modern-Era Retrospective analysis for Research and Applications version 2 (MERRA-2, replay) |
| **Stratospheric ozone chemistry** | UCX[d] | GMI standard stratospheric chemistry[e] |
| **Anthropogenic Emissions** | Community Emissions Data System (CEDS)[f] | MACC/CityZEN EU projects (MACCity) + RCP8.5[e] |
| **Biomass burning Emissions** | Global Fire Emissions Database version 4s (GFED4s)[g] | Global Fire Emissions Database version 4s (GFED4s)[g] |
| **Biogenic VOC Emissions** | Model of Emissions of Gases and Aerosols from Nature version 2.1 (MEGAN)[h] | Model of Emissions of Gases and Aerosols from Nature version 2.1 (MEGAN)[h] |

[a]Replay simulation of NASA Goddard Earth Observing System (GEOS) coupled to the Global Model Initiative (GMI) chemical mechanism and meteorological information from MERRA-2 reanalysis data. At each time step, the model inputs 3-

hourly averaged MERRA-2 meteorology output (zonal and meridional winds, temperature, pressure), which is used to adjust the model toward the MERRA-2 reanalysis (Orbe et al., 2017; Liu et al., 2020).

[b]DOI: 10.5281/zenodo.3974569

[c]https://earth.gsfc.nasa.gov/acd/models/gmi/models

[d]Universal tropospheric-stratospheric Chemistry eXtension, which combines both tropospheric and stratospheric reactions into a single chemistry mechanism.

[e]Rotman et al. (2004)

[f]Hoesly et al. (2018); CEDS provides monthly average anthropogenic emissions at the 0.5°x0.5° resolution using previously existing emissions inventories.

[g]Giglio et al. (2013) after 1997; prior to 1997, estimated using a GFED4s climatology with interannual variability imposed using scale factors from the Total Ozone Mapping Spectrometer aerosol index as in Duncan et al. (2003); monthly 0.25° resolution.

[h]MEGANv2.1 with updates from Guenther et al. (2012). Biogenic VOC emissions are calculated depending on the emissions timestep (e.g.., hourly at 4°x5°, every 30 minutes for 2°x2.5° resolution).

### 2.2.1 GEOS-Chem

We used two simulations with GEOS-Chem version 12.9.3 (GC) (Bey et al., 2001) at different horizontal resolutions (GC 4x5 and GC 2x2.5; DOI: 10.5281/zenodo.3974569) in this analysis (Table 2). Both simulations, using the native 72 vertical pressure levels, were carried out from 1980-2017 driven by reanalysis data from the Modern-Era Retrospective analysis for Research and Applications version 2 (MERRA-2) (Gelaro et al., 2017) developed by the NASA Global Modeling and Assimilation Office (GMAO). We used a 10-year spin-up simulation at 4°x5° for initialization. GEOS-Chem includes detailed $HO_x$-$NO_x$-VOC-ozone-$BrO_x$-aerosol tropospheric chemistry with over 200 species, and this version includes updated halogen (Wang et al., 2019) and isoprene chemistry (Bates and Jacob, 2019). Emissions were computed by the Harvard-NASA Emissions Component (HEMCO) (Keller et al., 2014) and were the same in both simulations. The global anthropogenic emissions inventory was the Community Emissions Data System (CEDS) (Hoesly et al., 2018), provided at a monthly 0.5° x 0.5° resolution. The CEDS inventory improved upon other inventories by using a consistent methodology for all emissions sectors, updated emission factors, and updated scaling inventories (Hoesly et al., 2018; McDuffie et al., 2020). Biogenic VOC emissions were calculated at each emissions timestep (e.g., hourly at 4°x5°, every 30 minutes at 2°x2.5°) by the Model of Emissions of Gases and Aerosols from Nature version 2.1 (MEGAN) with meteorological inputs from MERRA-2 (Guenther et al., 2012). Biomass burning emissions were provided via the monthly Global Fire Emissions Database (GFED) version 4s for 1997 and onward (Giglio et al., 2013). Before 1997, biomass burning emissions were estimated using a GFED4s climatology with interannual variability imposed using scale factors from the Total Ozone Mapping Spectrometer (TOMS)

aerosol index (Duncan, 2003). Biogenic soil $NO_x$ emissions were calculated online (Hudman et al., 2012). Lightning $NO_x$ emissions were constrained at ~6 Tg N per year and distributed to match satellite climatological observations of lightning flashes while maintaining coupling to deep convection from meteorological fields (Murray et al., 2012). Monthly mean methane concentrations were prescribed in the model surface layer from interpolation of the long term NOAA ESRL GMD flask observations (Murray, 2016). We used the Universal tropospheric-stratospheric Chemistry eXtension (UCX) to represent stratospheric chemistry in both simulations, which combined both stratospheric and tropospheric reactions into a single chemistry mechanism (Eastham et al., 2014). This differs from the linearized ozone (Linoz) mechanism (McLinden et al., 2000), which is frequently used in GEOS-Chem applications and calculates the evolution of most stratospheric species offline via archived monthly mean production rates and loss frequencies. While computationally efficient, the simplifications in Linoz may have consequences for STE. Using UCX allowed for a better representation of the stratosphere. We saved out 3-hourly averaged 72-layer 3D profiles for all GEOS-Chem species, resulting in >2.5 TB of model data in the 4°x5° and ~8 TB in the 2°x2.5° simulations for 1980-2017.

We performed two sensitivity tests at the coarse (4°x5°) resolution due to computational constraints. One simulation held anthropogenic emissions constant throughout 1980-2017. Note that only anthropogenic emissions in the CEDS inventory are held constant (e.g., $NO_x$, $SO_2$, CO, $NH_3$, NMVOCs, black carbon, and organic carbon). The other simulation held the meteorological condition as 1980 with varying anthropogenic emissions. These sensitivity tests allowed us to examine the impact of emissions and meteorology on the tropospheric ozone trend. Further, we used an earlier GEOS-Chem simulation (v10-01; http://wiki.seas.harvard.edu/geos-chem/index.php/GEOS-Chem_versions#GEOS-Chem_10_release_series) at 4°x5° for 1980-2010 described by Hu et al. (2017) as a supplemental analysis. Some major differences relevant to the ozone trend in this early simulation include 1) the MERRA reanalysis meteorological data (Rienecker et al., 2011), 2) A simplified linearized stratospheric chemistry and cross-tropopause ozone fluxes (Linoz; McLinden et al., 2000), 3) 47 vertical pressure levels, and 4) global anthropogenic emissions (decadal resolution and interpolated to a yearly basis) and biomass burning emissions (monthly resolution) from the MACCity inventory prior to 2005 and based on the Representative Concentration Pathway (RCP) 8.5 emissions scenario after (Granier et al., 2011). This earlier simulation version helps us to interpret low ozone biases in the recent GEOS-Chem version (Sect. 4.3).

### 2.2.2 NASA MERRA2-GMI

We also used a replay simulation from 1980-2017 of the NASA GEOS GMI, which uses the GEOS version 5 global atmospheric general circulation model (Molod et al., 2015) coupled with the GMI chemical mechanism (Nielsen et al., 2017) (http://acd-ext.gsfc.nasa.gov/Projects/GEOSCCM/MERRA2GMI). It includes a complete treatment of stratospheric and tropospheric chemistry and uses the Goddard Chemistry Aerosol Radiation and Transport (GOCART) module for aerosols. The simulation was run at c180 on the cubed-sphere, which is ~50 km horizontal resolution, and output on the same

0.5°x0.625° (latitude x longitude) grid as MERRA-2. The model was run in replay mode, which is described in detail in Orbe
et al. (2017). Briefly, the model initially runs forward in a free state and is compared to the 3-hourly averaged core MERRA-
2 meteorological fields (zonal and meridional winds, temperature, pressure). The difference is evaluated and the model
rewound, running forward with the added increment at each time step needed to adjust the model meteorology toward the
MERRA-2 reanalysis (Orbe et al., 2017; Liu et al., 2020). Anthropogenic emissions were provided by MACCity (Granier et
al., 2011) until 2010, then derived using the RCP 8.5 scenario after. Biomass burning emissions were calculated using the
GFED4s coupled with pre-1997 interannual variability, using the same methodology described above. Biogenic emissions
were provided by MEGANv2.1 (Guenther et al., 2012). MERRA2-GMI has been used previously to investigate both
tropospheric and stratospheric ozone and has been shown to capture the diurnal cycle of ozone, the relationship between ozone
and temperature during summertime, and trends in tropospheric $NO_2$ as observed remotely by OMI (Strode et al., 2019; Kerr
et al., 2019) that aid in explaining global ozone trends (Ziemke et al., 2019).

Additionally, the MERRA2-GMI simulation contains a stratospheric ozone tracer (STO3) to diagnose stratospheric ozone
intrusion in the troposphere, which influences tropospheric ozone trends and interannual variability (Ordóñez et al., 2007; Liu
et al., 2020). This tracer, which has no sources in the troposphere, was set equal to simulated stratospheric ozone flux at the
tropopause, as determined by the artificial tracer, e90, introduced by Prather et al. (2011). STO3 was then transported through
the troposphere and removed using chemical loss rates and surface deposition fluxes run online at each time step from the full
chemistry simulation. MERRA2-GMI produces a credible stratospheric transport circulation (Orbe et al., 2017), which agrees
with observations for trends in the upper troposphere and lower stratosphere (Wargan et al., 2017, 2018). STO3 has been used
to explain recent observed decreases in lower stratospheric ozone over the Northern Hemisphere and extratropics (Orbe et al.,
2020; Wargan et al., 2018), as well as the influence of stratospheric ozone on the interannual variability in tropospheric ozone
over North America and Europe (Liu et al., 2020). Here, we used the STO3 tracer to explore the influence of STE on
tropospheric ozone trends.

### 2.2.3 Model-measurement evaluation

To avoid biases in our model-measurement evaluation resulting from averaging model output prior to sampling, each model
was sampled to match ozonesonde launch locations and times as closely as possible. Model ozone output was saved as 3-hour
averages, and each model was sampled to match ozonesonde launch times paired to the closest 3-hour timestamp. Each
individual ozonesonde profile was used to calculate trends. Both GC simulations and MERRA2-GMI were also sampled at
surface site locations provided by TOAR. Only daytime ozone values were used (between 8 and 20 h local time), following
the definition used in the TOAR Surface Ozone Database. Further, each surface site was sampled in the model at the pressure
level most closely matching the site's elevation, which was converted to pressure assuming a standard atmosphere. Surface

daytime ozone concentrations were then averaged monthly for the analysis. All model trends were calculated using the same methods as the observational trends.

## 3 Observational evidence for global ozone increases

### 3.1 Validation of ozonesonde trends with surface observations

Regular ozonesonde launches at long-term sites during certain days of the week or month represent untargeted sampling that allows for a systematic characterization of the vertical distribution of the entire troposphere and above. However, concerns about the suitability of ozonesondes for long-term trend analyses have been raised previously (Saunois et al., 2012; Liu et al., 2016). The concern is that ozonesondes launched only a few times per month capture snapshots of ozone changes over time and may not fully capture trends. By contrast, ozone is measured continuously on an hourly basis at the surface sites, making

it likely that these sites capture robust trends in long-term data, though they reflect only the trends in the atmospheric boundary layer. To assess the ability of our ozonesonde sites launching at least 3 times per month to accurately represent overall trends, we compared the lowest reliably available pressure level (800 hPa) to co-located surface TOAR sites within a 100 km radius. The 800-hPa pressure level is typically within the atmospheric boundary layer and should be mostly affected by similar processes as the surface sites.

In most seasons, we found that trends from the surface sites and the ozonesondes correlate significantly (r>0.5, p<0.1 for all), while wintertime often shows the worst agreement due to a lower boundary layer height. Summer is typically when trends match most closely, as the boundary layer is deepest then. At all five co-located sites during summer (Boulder (USA), Hohenpeissenberg (Europe), Payerne (Europe), Uccle (Europe), and Tateno (Japan)), trends between the surface and 800 hPa

match in terms of direction, and the magnitude of trends differ by <30% (Fig. S3). This suggests that ozonesondes launching at least 3 times per month are able to capture long-term seasonal trends. The absolute values of ozone can differ widely between measurement techniques, with surface sites being systematically lower due to the increased influence of dry deposition (Travis and Jacob, 2019). Previous work has typically used ozonesonde data that launch 4 times per month (Lu et al., 2019). However, along with our other data requirements, this restriction would limit the number of sites to just 15, eliminating nearly all

Southern Hemisphere and polar sites and negatively impacting our global analysis. Here, we show that trends in low-level ozonesondes and TOAR sites largely match each other, and we conclude that we are able to use the ozonesonde sites launching at least 3 times per month to understand trends throughout the vertical column.

### 3.2 Free tropospheric ozone trends

Trends in ozonesonde data suggest tropospheric ozone has increased throughout the troposphere since the 1980s and 1990s. Of the 25 ozonesonde stations examined globally from 1990-2017, 14 show statistically significant increases from 800 to 400

hPa (Fig. 3). We caution that these results are derived from both homogenized and non-homogenized data depending on availability (see Table 1 for a list of homogenized sites). The impact of homogenization is shown in Fig. S4, with homogenization affecting trend magnitudes but rarely the sign of the trends compared to non-homogenized data. Across all pressure levels, these 14 sites average an increase of $1.8 \pm 1.3$ ppb decade$^{-1}$ ($3.5\% \pm 2.6\%$ decade$^{-1}$), ranging from 0.1 to 5.3 ppb decade$^{-1}$ (0.2 to 10.6% decade$^{-1}$) since the 1990s. At the 9 sites that have records from 1980, 5 show consistent increases averaging $1.3 \pm 0.7$ ppb decade$^{-1}$ ($2.6\% \pm 1.4\%$ decade$^{-1}$) and ranging from 0.1 to 3.0 ppb decade$^{-1}$ (0.1 to 5.9% decade$^{-1}$) (Fig. S5). Over half of all ozonesonde sites from 1990-2017 show increasing ozone in the free troposphere (700-400 hPa) at an average rate of $1.9 \pm 1.3$ ppb decade$^{-1}$ ($3.6\% \pm 2.4\%$ decade$^{-1}$), but trends range widely, from 0.1 to 5.3 ppb decade$^{-1}$ (0.1 to 9.9% decade$^{-1}$). While relative trends (taken relative to the mean ozone concentration at each pressure level from 1990-2017) are remarkably constant through the troposphere at most sites, they tend to be larger closer to the surface (4.3% decade$^{-1}$ below 700 hPa on average, compared to 3.5% decade$^{-1}$ above 700 hPa), reflecting the importance of emissions changes on ozone trends (Fig. S6). Trends at sites that are not increasing show mostly insignificant decreasing trends, with few showing statistically significant decreases. The only records with strongly negative trends are the lower troposphere at Wallops Island (eastern US, $-2.2 \pm 0.6$ ppb decade$^{-1}$), the upper troposphere at Macquarie Island (Southern Ocean, $-1.2 \pm 0.4$ ppb decade$^{-1}$), the lower troposphere at Broadmeadows (south-eastern Australia, $-1.2 \pm 0.2$ ppb decade$^{-1}$), and the extreme upper troposphere at Eureka (Polar Canada, $-2.7 \pm 1.6$ ppb decade$^{-1}$).

The strongest increasing trends from 1990-2017 occur in Japan, averaging $3.8 \pm 0.8$ ppb decade$^{-1}$ ($7.1\% \pm 1.5\%$ decade$^{-1}$) across all pressure levels and ranging from 2.4 to 5.3 ppb decade$^{-1}$ (4.4% to 9.9% decade$^{-1}$). Caution should be taken to not over-interpret the Japanese trends, as a potential step change occurs at these sites around 2010 in the troposphere (Fig. S1). While this may be partially due to a change in sensor response, these step changes are not visible in the stratosphere (Fig. S1), suggesting that these trends mostly reflect the rapid increase in emissions over Asia in the past 4 decades. Similarly, all NH Polar sites except Eureka show increasing trends, averaging $1.6 \pm 0.9$ ppb decade$^{-1}$ ($3.1\% \pm 1.7\%$ decade$^{-1}$; ranging from 0.4 to 3.3 ppb decade$^{-1}$). Over North America, the 2 Canadian sites (Edmonton and Goose Bay) show consistent increases throughout the tropospheric column, averaging $1.3 \pm 0.9$ ppb decade$^{-1}$ ($2.5\% \pm 1.7\%$ decade$^{-1}$; ranging from 0.5 to 3.3 ppb decade$^{-1}$). Half of sites over Europe also show increasing trends, averaging $1.9 \pm 1.1$ ppb decade$^{-1}$ ($3.4\% \pm 2.0\%$ decade$^{-1}$; ranging from 0.1 to 4.3 ppb decade$^{-1}$). Over the Southern Hemisphere, 2 of the 6 sites show smaller increasing trends from 1990-2017 compared to other regions, averaging $0.7 \pm 0.6$ ppb decade$^{-1}$ ($2.1\% \pm 1.8\%$ decade$^{-1}$; ranging from 0.3 to 1.7 ppb decade$^{-1}$). Hilo (Hawaii) in the tropics shows insignificant trends below 600 hPa, averaging $0.6 \pm 0.7$ ppb decade$^{-1}$ ($1.3\% \pm 1.5\%$ decade$^{-1}$), and insignificant increases above 600 hPa, averaging $1.1 \pm 0.7$ ppb decade$^{-1}$ ($2.2\% \pm 1.4\%$ decade$^{-1}$).

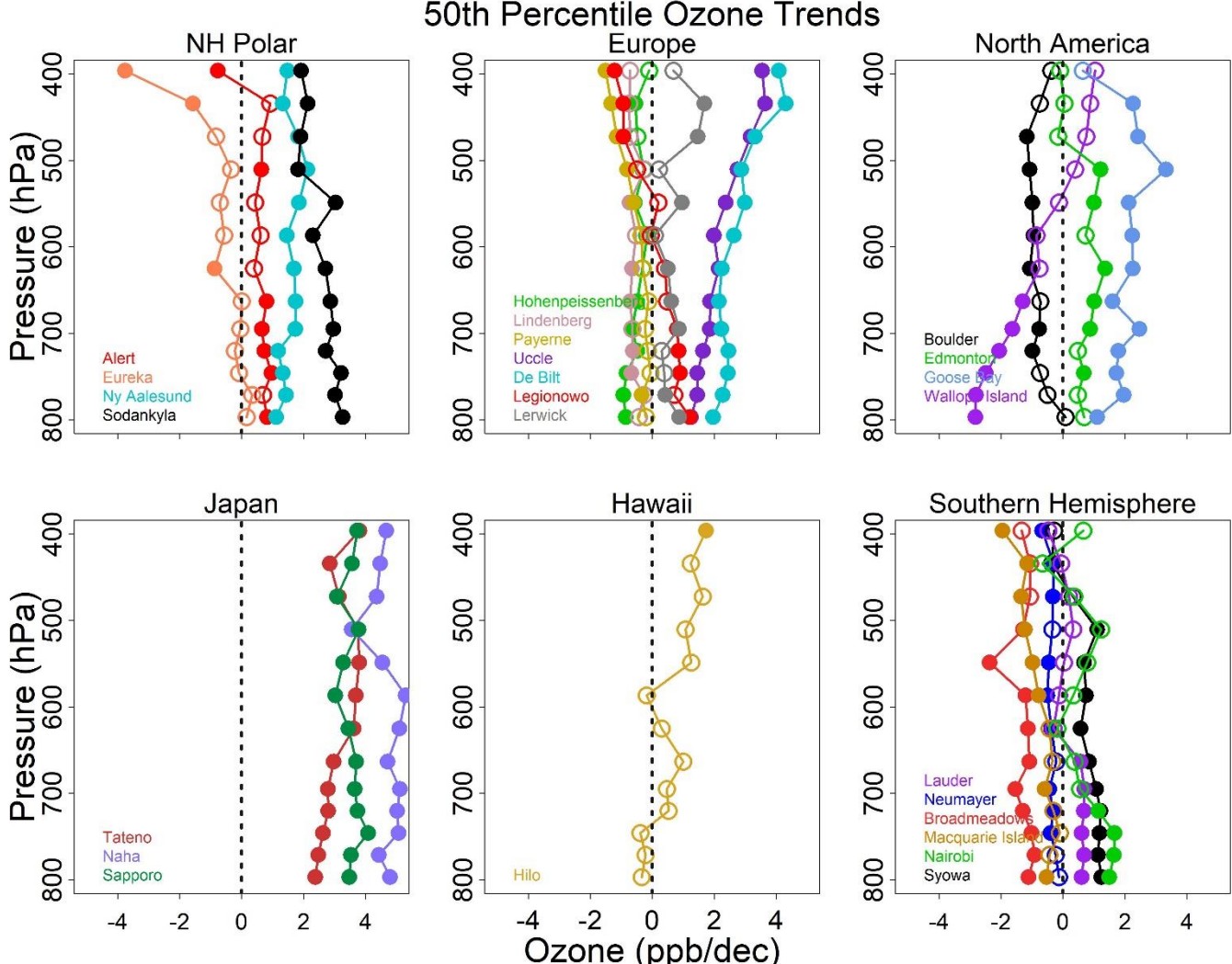

**Figure 3. Trends (ppb decade$^{-1}$) through the free troposphere (800-400 hPa, reduced to GEOS-Chem pressure levels) at the 25 global ozonesonde sites with data from 1990-2017, distributed into six regions. Solid circles indicate that the trends are statistically significant (p<0.1), while open circles denote statistically insignificant trends.**

Figure 4 depicts the shift in overall ozone distributions at all pressure levels between the first (1990-1994) and last (2013-2017) 5 years of the time series with all sites grouped into five of the six regions (i.e., all except Hawaii). In each region, distributions from 800-400 hPa shift in a positive direction, with increases in medians averaging 2.5 ppb globally and ranging up to 3.5 ppb over Japan. Across all sites in these regions, the largest absolute and relative shifts occur in the lower troposphere (>700 hPa). Changes in medians average 2.2 ppb (5.1%) in the lower troposphere and 1.3 ppb in the free troposphere (2.6%).

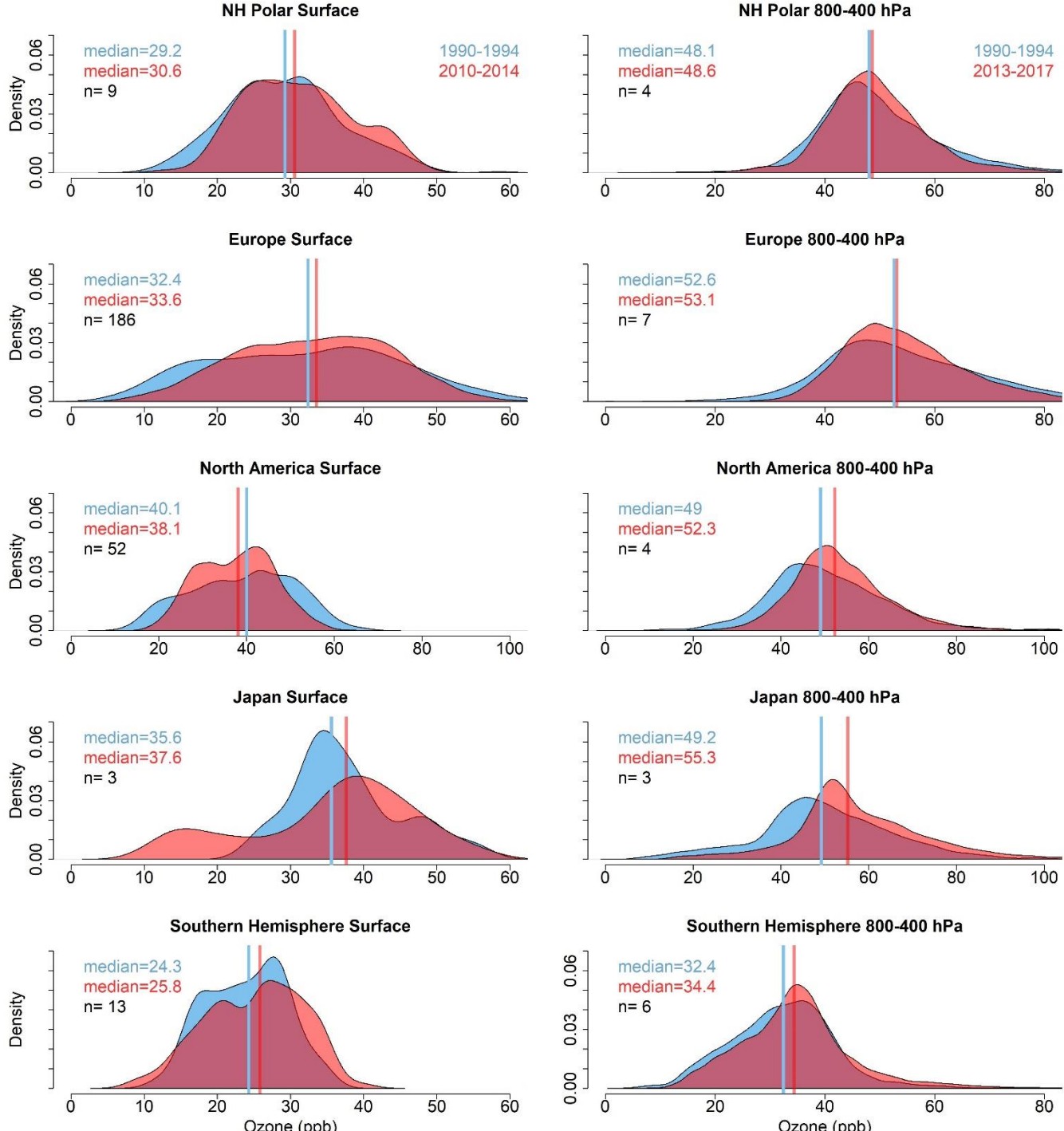

**Figure 4. Changes in ozone concentration (ppb) distributions between the first five years of analysis (red; 1990-1994) and the last five years of analysis (blue; 2010-2014 for surface; 2013-2017 for sondes) shown as density functions at the surface (background sites compiled by TOAR) and throughout the troposphere (all ozone values measured by ozonesondes in the pressure range 800 to 400 hPa). Median concentrations are shown with vertical lines, and the corresponding values and number of sites are recorded inset.**

The generally increasing ozone concentrations measured by ozonesondes are consistent with satellite and aircraft data. Satellite measurements from the Aura Ozone Monitoring Instrument/Microwave Limb Sounder (OMI/MLS) from 2005-2016 show widespread increases of ozone across the tropics and mid-latitudes, ranging up to >3 DU decade$^{-1}$ over Asia (Ziemke et al., 2019). This finding is corroborated by global chemistry climate models, which indicate that the tropospheric ozone burden has increased since 1990 (Myhre et al., 2017; Ziemke et al., 2019). In the simulations used in this work, we also find that the ozone burden has increased since 1980, which we discuss further in Sect. 4.3. Free tropospheric and tropospheric column ozone measured by IAGOS also suggests that ozone has increased across the Northern Hemisphere since the 1990s (Gaudel et al., 2020; Petzold et al., 2015; Cohen et al., 2018) at an average rate of 2 ppb decade$^{-1}$, which agrees with the average 2.0 ± 1.3 ppb decade$^{-1}$ increase in Northern Hemisphere FT ozonesonde measurements. Although some variation is expected when comparing regions to individual sonde launch locations, our results show good agreement with previous analyses of FT ozone (700-300 hPa) since the 1990s using IAGOS flight data. Over Europe, Gaudel et al. (2020) found an increasing trend of 1.3 ± 0.2 ppb decade$^{-1}$, slightly lower than our result of 1.9 ± 1.1 ppb decade$^{-1}$ but within uncertainty. Gaudel et al. (2020) report an increase of 1.3 ± 0.9 over the Southeast US FT, which aligns with our findings in the upper troposphere at Wallops Island (Virginia, US; 0.8 ± 0.3 ppb/dec). Over Eastern North America, an increase of 1.7 ± 0.4 ppb decade$^{-1}$ is in good agreement with ozonesonde measurements at Goose Bay (Eastern Canada; 2.0 ± 0.7 ppb/dec). This remarkable agreement between ozonesondes and other measurement platforms lends further evidence that ozonesondes launching 3 times per month are able to capture long-term trends in tropospheric ozone.

There is much discussion about the number of profiles needed for statistical analyses of global ozone trends, and recent studies have suggested that 14 profiles per month are needed (Chang et al., 2020). However, this number of profiles is not possible under the current ozonesonde sampling landscape. Here, we show that careful selection and treatment of ozonesonde data can lend important insights to global ozone trends that are highly vertically resolved. We note that these trends may not be considered globally representative, but rather they offer an additional insight into ozone changes over the past few decades. That we find good agreement between ozonesonde trends and trends from other data sources suggests that ozonesonde information is an important part of the ozone monitoring landscape in determining global trends.

It is important to note that we have not performed a seasonal analysis of ozonesonde data. Analyses of ozonesonde sites in the tropics point to the seasonal variability of ozone and show that trends are driven primarily by changes during certain months. For example, Thompson et al. (2021) did not find significant trends at Nairobi, Kenya, on an annual basis (consistent with our results), but found that FT ozone increased during February-April by 5-10% decade$^{-1}$ while decreasing during August-September. Other tropical sites show similar patterns – annual trends are insignificant, while seasonal trends are much larger. A seasonal analysis is beyond the scope of this paper, as the 3-4 launches per month may not give enough information for

robust monthly or seasonal trend analysis. Future investigations of ozone trends should consider the impact of specific months

or seasons, provided it can be done in a statistically meaningful way, to aid in identifying drivers of trends.

### 3.3 Surface baseline ozone trends

While surface ozone trends have been discussed in previous analyses (Gaudel et al., 2018; Cooper et al., 2020), we focus specifically on daytime ozone trends rather than on trends in monthly mean ozone that average all times of day (Parrish et al.,

2014); we also consider a greater number of sites covering a larger geographical area than other studies attempting to characterize baseline ozone. Specifically, we include 271 sites, including additional sites in the poorly sampled Southern Hemisphere, while restricting site locations to rural background areas.

Despite regional decreases over the US and Europe, surface ozone increases in most places globally since the 1990s, ozone

distributions have generally shifted up across the timeframe, and medians have largely increased (Fig. 4). At sites outside of the United States and Europe at low elevations, 73% show increasing trends (24 of 33 sites). Including the United States and Europe sites, we find that 42% of global surface background sites (114 of 271) show ozone increases since the 1990s, with notable decreases at 48 of the 52 United States sites and 100 of 186 Europe sites due to emissions regulations (Fig. 5). Surface ozone changes at individual sites globally range from -7.5 to +5.2 ppb decade$^{-1}$. Across all sites, increases average 1.0 ± 0.8

530 ppb decade$^{-1}$, and decreases average -1.4 ± 1.2 ppb decade$^{-1}$. For sites outside of the US and Europe, increases average 1.4 ± 0.9 ppb decade$^{-1}$, and decreases average -1.3 ± 0.8 ppb decade$^{-1}$, with the largest increases occurring over Asia. Decreases in eastern China (LinAn) can be attributed to the prevalence of clean marine air masses impacting that site during fall that do not reflect the growing urban emissions in China (Xu et al., 2008). Our results are consistent with other global analyses of surface ozone data that have shown increases over varying timeframes beginning in the 1990s at far fewer sites spanning a narrower

slice of the globe (Cooper et al., 2020, 2014). Of the expanded 258 sites in the Northern Hemisphere analyzed here, we find increases at 103 sites (40%), ranging from <0.1 to 5.2 ppb decade$^{-1}$. Focusing on Northern Hemisphere trends outside of the United States and Europe, we find increasing trends at 13 of the 20 sites (65%), averaging 1.4 ± 0.9 ppb decade$^{-1}$ (0.5 to 5.2 ppb decade$^{-1}$). At the 13 Southern Hemisphere sites analyzed here, we find increases at 11 sites (85%) since 1990, ranging from 0.5 to 1.8 ppb decade$^{-1}$. Our results are consistent with findings from Cooper et al. (2020), who found that about half of

Northern Hemisphere sites with significant trends (5 out of 10 sites) show increasing trends ranging from 0.7-1.7 ppb decade$^{-1}$ and 71% of Southern Hemisphere sites (5 out of 7) show increasing trends (0.3 to 1.5 ppb decade$^{-1}$). Increases at surface sites are shown in Fig. 4, where the medians of all distributions except North America have shifted in a positive direction from the first 5 years of analysis (1990-1994) to the last 5 years (2010-2014), with changes in median ozone concentration across all regions averaging 2.0 ppb (6.0%) at the surface, closely matching overall ozone increases at the 27 sites observed globally in

Cooper et al. (2020) since 1995 (1.6 ppb).

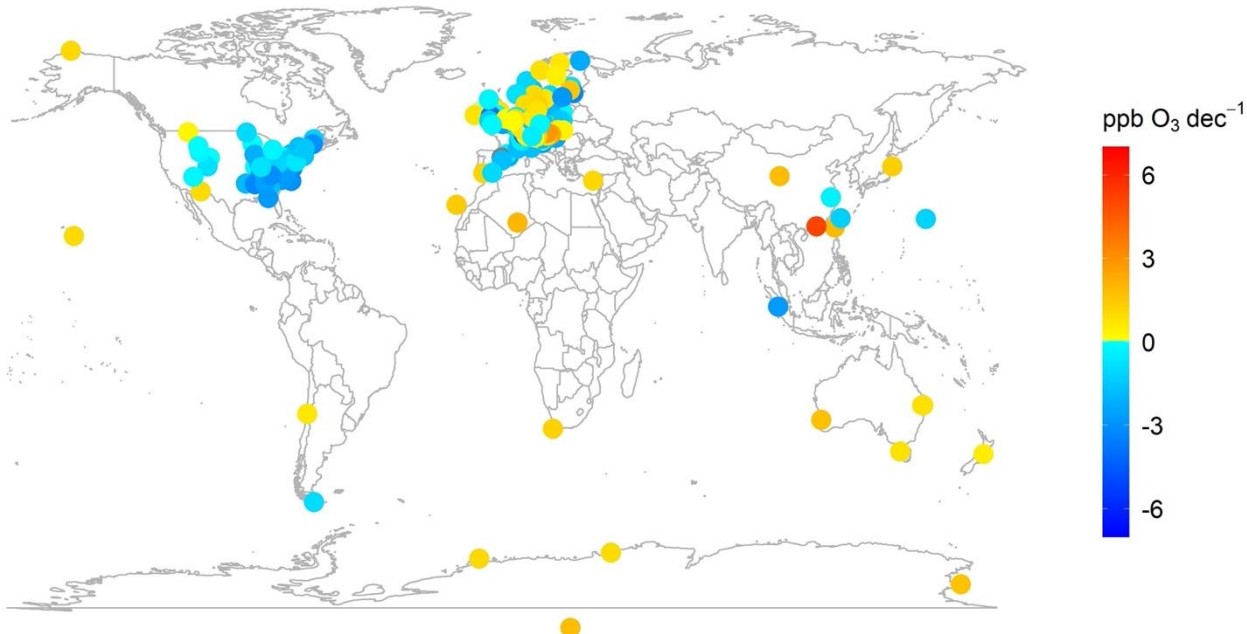

**Figure 5. 1990-2014 daytime surface ozone trends (ppb decade$^{-1}$) at sites compiled in the TOAR database. Warm colors indicate increasing trends, and cool colors indicate decreasing trends.**

### 3.4 Trends at high elevation sites

High elevation surface sites provide another line of evidence regarding regional baseline ozone (e.g., ozone that is not influenced by local emissions) trends, as they are regionally representative of the lower troposphere (Logan et al., 2012; Cooper et al., 2014; Parrish et al., 2012, 2014). Careful filtering of data at some of these high elevation sites can also isolate the influence of lower FT air (Cooper et al., 2020; Lin et al., 2014). Analyses of high elevation sites have focused primarily on Europe (Cooper et al., 2020; Logan et al., 2012), although a limited number of sites in North America, Japan, Hawaii, and China have also been studied (Parrish et al., 2012, 2014; Cooper et al., 2014; Lin et al., 2014; Xu et al., 2016). Here, we do not attempt to recalculate trends at these sites, but rather examine previously reported trends and compare them to lower and free tropospheric ozonesonde trends. We show that the trends measured by ozonesondes match those of high elevation and mountaintop surface trends in most locations, adding confidence to the trends we derive from ozonesondes launching at least 3 times per week.

Two mountaintop sites influenced by FT air are Mauna Loa (Hawaii) and Mt. Waliguan (China). At both of these sites, FT trends measured at the mountaintop sites show increasing FT (700-400 hPa) ozone trends. At Mt. Waliguan, FT trends can be isolated using nighttime ozone values, and measurements show an increase in FT ozone of $2.8 \pm 1.6$ ppb decade$^{-1}$ from 1994-2013 (Xu et al., 2016) and $1.7 \pm 0.5$ ppb decade$^{-1}$ from 1994-2016 (Cooper et al., 2020). This finding is attributed to both transport of increasing anthropogenic emissions and intensifying STE, which can explain 60% of the springtime ozone increase (Xu et al., 2016, 2018). While we do not analyze any ozonesonde launch locations over China and therefore do not have a direct comparison to sonde information, it is important to recognize the pattern of increasing FT ozone at multiple sites throughout the globe. At Mauna Loa, the influence of FT air can be isolated under nighttime conditions with low relative humidity. Cooper et al. (2020) found that FT ozone at Mauna Loa has increased by $2.4 \pm 1.0$ ppb decade$^{-1}$ since 1995. Annual trends from 1991-2010 were found to be $3.1 \pm 0.7$ ppb decade$^{-1}$ (Oltmans et al., 2013), driven by increasing autumn trends ($3.5 \pm 1.4$ ppb decade$^{-1}$; 1980-2012) (Lin et al., 2014). The trend reported in Cooper et al. (2020), which best matches our analysis timeframe, is higher than the average FT trends we calculated over Hilo from ozonesonde measurements from 1990-2017 ($0.9 \pm 0.6$ ppb decade$^{-1}$ from 700-400 hPa), but falls within the range measured in the FT (range of -0.2 to 1.7 ppb decade$^{-1}$).

The other high elevation sites have been found to be more representative of regional ozone trends in the lower troposphere than the FT. Two European mountaintop sites (Zugspitze, Sonnblick) show decreasing trends since 1995, $-0.8 \pm 0.6$ and $-1.0 \pm 0.7$ ppb decade$^{-1}$, respectively, while a third mountaintop site, Jungfraujoch, exhibits an insignificant trend of $0.2 \pm 0.6$ ppb decade$^{-1}$ at night (Cooper et al., 2020). We find good agreement between closely located sonde and mountaintop trends. Both Zugspitze and Sonnblick are closely located to the Hohenpeissenberg ozonesonde location (within 100 km), which shows a decreasing trend of $-0.8 \pm 0.2$ ppb decade$^{-1}$ in the lower troposphere, within the range of trends reported for Zugspitze and Sonnblick. Jungfraujoch is near the Payerne ozonesonde location (within 100 km) and shows an insignificant decreasing trend of $-0.2 \pm 0.1$ ppb decade$^{-1}$, which overlaps with the trend reported at Jungfraujoch. A consistent picture is difficult to put together for all of Europe considering the large variation in local trends, but overall our lower tropospheric ozone trends from sonde data encompass those found at mountaintop sites.

Over the United States from 1995-2017, Cooper et al. (2020) reported on two lower tropospheric high elevation sites, Centennial (WY) and Gothic (CO). Trends are $-1.5 \pm 1.2$ ppb decade$^{-1}$ and $-1.9 \pm 0.8$ ppb decade$^{-1}$ during the daytime, respectively. At the Boulder (CO) ozonesonde measurements, lower tropospheric trends average $-0.5 \pm 0.4$ ppb decade$^{-1}$, agreeing with surface trends. Over the South Pole, only 24-hour trends from 1995-2018 were reported by Cooper et al. (2020) due to the lack of a diurnal ozone cycle; these averaged $1.5 \pm 0.6$ ppb decade$^{-1}$. We find a consistent trend of $1.2 \pm 0.1$ ppb decade$^{-1}$ in the lower troposphere at Syowa Station (coastal Antarctica). Differences in the increases at these two stations may occur as a function of station location and whether anthropogenic sources or meteorological variables are the main drivers of ozone trends at each station. At the South Pole, increases are associated with ozone-rich air from the upper troposphere and

lower stratosphere, whereas Syowa, located at 69° S, is primarily impacted by marine air and air-mass transport from regions near South America (Kumar et al., 2021). It is also important to note that Syowa switched sensors from CI to ECC in 2010, which could impact trends.

## 3.5 Potential drivers of observed ozone change

Across all regions, we find that increases in 5[th] percentile ozone at the surface and aloft since the 1990s contribute to increases in median ozone. To estimate 5[th] percentile ozone trends, we first calculated the 5[th] percentile ozone in each month at each pressure level for all individual sites, then used the quantile regression method to calculate the trends of the 5[th] percentiles of

measurements year-round. Figure 6 shows the trend of 5[th] percentile ozone across ozonesonde and surface sites grouped into the six regions. At most locations globally (178 of 271 surface sites and 13 of 25 sonde sites), 5[th] percentile ozone has increased in both ozonesonde and surface trends, averaging $1.8 \pm 1.3$ ppb decade[-1], with 59% of those sites showing increases of greater than 1.0 ppb decade[-1] and ranging up to 4.9 ppb decade[-1] at surface sites and 5.6 ppb decade[-1] at ozonesonde sites. Notably, while 5[th] percentile surface ozone has increased significantly in the United States and Europe, peak surface ozone values

decreased in recent years (Fig. 4), reflecting reductions in regional anthropogenic emissions of ozone precursors (Yan et al., 2018a, b). In contrast, in the FT over Japan and the Southern Hemisphere, the entire ozone distribution has shifted higher. Over Japan, these increases have been attributed to transport from the Asian continent and reduced $NO_x$ emissions leading to decreased titration of ozone (Akimoto et al., 2015). Over the Southern Hemisphere, these increases occur in response to changing precursor emissions and large-scale dynamics, including an expansion of the Hadley cycle which may allow more

stratospheric, ozone-rich air to enter the troposphere (Lu et al., 2019; Zeng et al., 2017; Cooper et al., 2020).

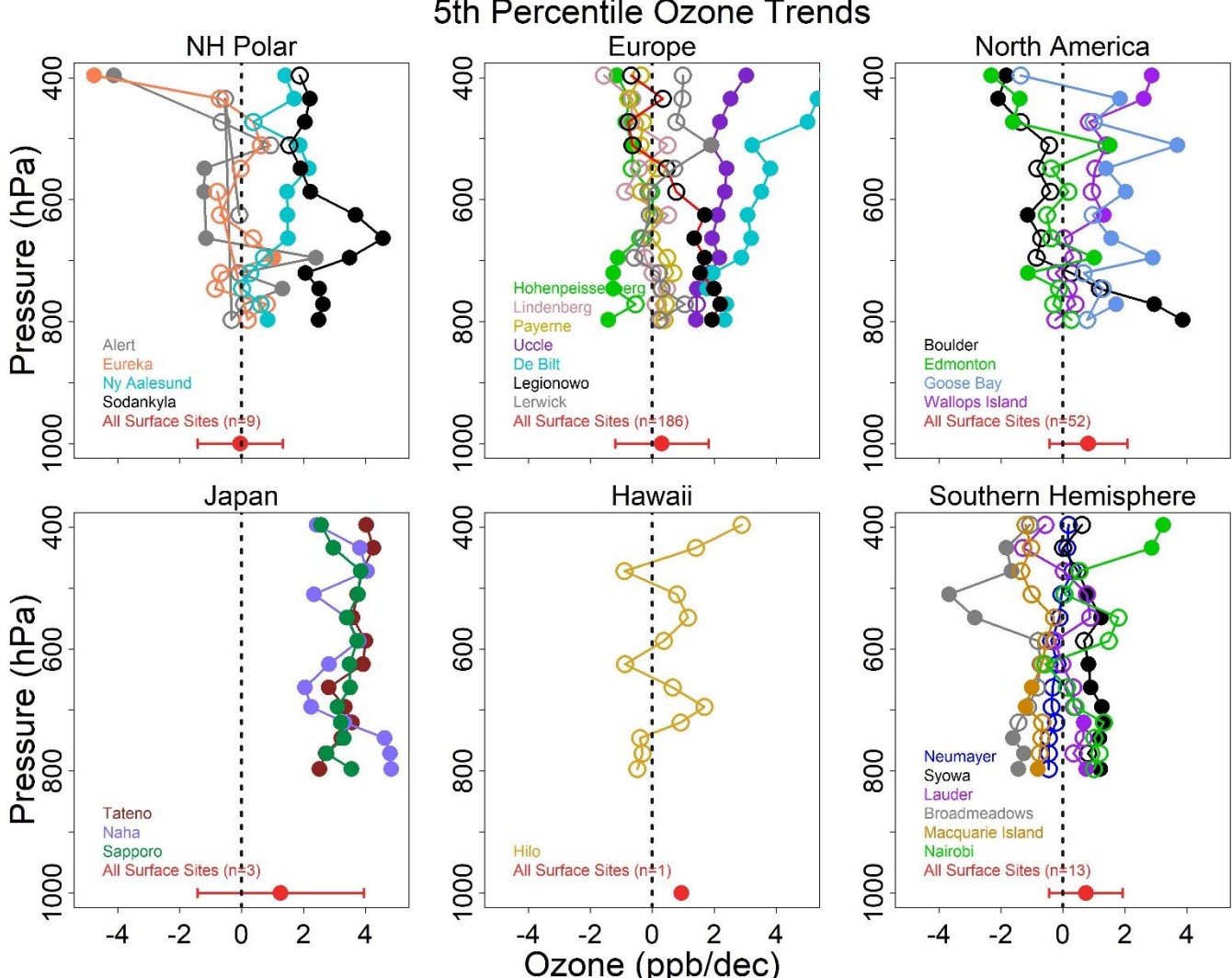

**Figure 6. Trends in 5th percentile ozone (ppb decade-1) through the free troposphere (800-400 hPa) at the 25 global ozonesonde sites**
**(1990-2017) and mean 5th percentile ozone trends at the surface for all 238 sites within the six designated regions (1990-2014). Solid**
**circles indicate that the trends are statistically significant (p<0.1), while open circles are statistically insignificant. Error bars for**
**surface background sites represents the standard deviation across all sites.**

Increasing 5th percentile concentrations are consistent with other analyses that suggest baseline ozone has been increasing, especially in the Northern Hemisphere. Increases in 5th percentile ozone have been attributed to a number of factors: decreased titration from NO$_x$ as a result of emissions decreases on a local scale, especially over urban areas in the United States and Europe (Yan et al., 2018b; Simon et al., 2015; Lin et al., 2017; Gao et al., 2013; Clifton et al., 2014), increases in methane concentrations (Lin et al., 2017), changes to large-scale processes such as STE (Parrish et al., 2012), and transport of ozone from the tropics and subtropics (Zhang et al., 2016; Gaudel et al., 2020). While all of these factors likely play a role in increased

5th percentile ozone in the Northern Hemisphere, multiple previous analyses suggest that regional, baseline ozone increases observed in rural locations with little impact from local emissions are best explained by transport from the tropics (Zhang et al., 2016, 2021). The largest emissions of ozone precursors have shifted toward low-latitude nations, especially in Southeast, East, and South Asia, where increased convection and temperature lead to more efficient ozone production compared to the mid-latitudes. This ozone is then transported poleward (Zhang et al., 2016). Tropospheric ozone increases in the middle

troposphere (550 to 350 hPa) over mid-latitudes can be largely explained in models through transport of ozone from low latitudes, with STE playing an important role in the upper troposphere (above 350 hPa) (Zhang et al., 2016; Gaudel et al., 2020). Only 15% of the ozone increase over the western US between 1980-2014 has been attributed to an increase in methane concentrations (Lin et al., 2017).

## 4 Models underestimate ozone trends

**4.1 Model reproduction of ozone trends in the FT**

We find that models comprising different resolutions, time-varying emissions, assimilated meteorological inputs, and chemical schemes tend to underestimate observed long-term ozone trends throughout the troposphere, and the direction of trends at some individual sites is not captured (Fig. 7). Across all 25 sites evaluated, the average 800-400 hPa observed ozone trend by ozonesondes is $0.8 \pm 1.7$ ppb decade$^{-1}$ from 1990-2017, reflecting the wide spread in observed trends, and the three simulations

underestimate this trend mostly in the Northern extratropics. Globally, MERRA2-GMI captures ~75% of the trend at $0.6 \pm 0.7$ ppb decade$^{-1}$, but both GC simulations drastically underestimate it and do not differ significantly between the different resolutions ($0.15 \pm 0.7$ ppb decade$^{-1}$ for the 4°x5° version; $0.1 \pm 0.9$ ppb decade$^{-1}$ for the 2°x2.5°). This result represents <20% of the overall average observational trend for both GC simulations from 1990-2017. Notably, MERRA2-GMI typically performs better in the upper free troposphere than the GC simulations in the northern mid-latitudes, matching 44% of the

observed trend from 600 to 450 hPa, while the GC simulations only capture 24%. An important note is that a notable step change occurred in MERRA2-GMI ozone after 1998 associated with an observing system upgrade incorporated into MERRA-2 (Stauffer et al., 2019). This step change impacts pressure levels mostly above our analysis range, and model ozone at pressures <450 hPa may be affected. MERRA2-GMI trends at pressures <450 hPa should thus be interpreted with caution. While the error bars do overlap between models and observations in all simulations, most of this error is due to regional

variability, and trends between models and measurements within regions often do not overlap.

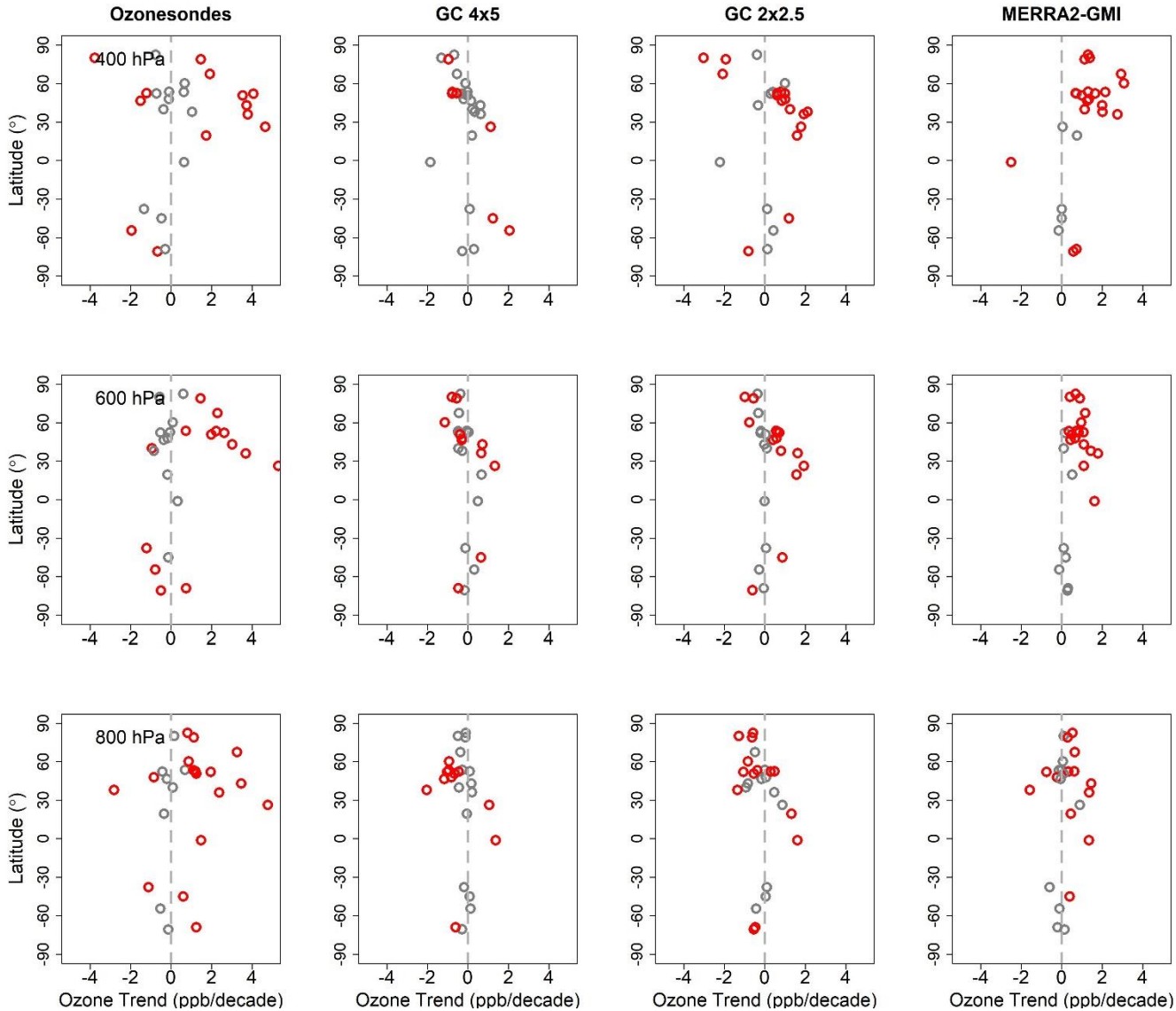

**Figure 7. Summary of 1990-2017 trends in ozonesondes (left column) and the simulations (other columns). GC 4x5 refers to the GEOS-Chem v12.9.3 simulations at 4°x5°, GC 2x2.5 is the same model at 2°x2.5°, and MERRA2-GMI refers to the NASA GEOS GMI at ~50 km resolution. The trend in ppb decade[-1] is plotted as a function of ozonesonde launch site latitude. Red circles indicate significant trends (p<0.1), and gray circles indicate insignificant trends.**

Figure 8 shows shifts in ozone distributions from 1990-2017 between 800 and 400 hPa. Most overall shifts in distribution from 800-400 hPa are captured by models in a qualitative sense, but shifts tend to be underestimated, most strongly by the GC simulations. Both GC simulations capture the observed increases in all regions except the NH Polar region and Europe, where the models both show decreasing trends in contrast to observations (Fig. 4; also shown in Fig. 7). The median ozone increases

are underestimated by an average of 3 ppb in both simulations. In contrast, MERRA2-GMI captures the observed increases everywhere, but underestimates these increases over North America by 0.9 ppb. MERRA2-GMI also overestimates the median

increase over Europe, Japan, and the NH Polar region by 1.6, 2.1, and 1.7 ppb, respectively, yet it captures within 0.5 ppb the overall median increases over the Southern Hemisphere.

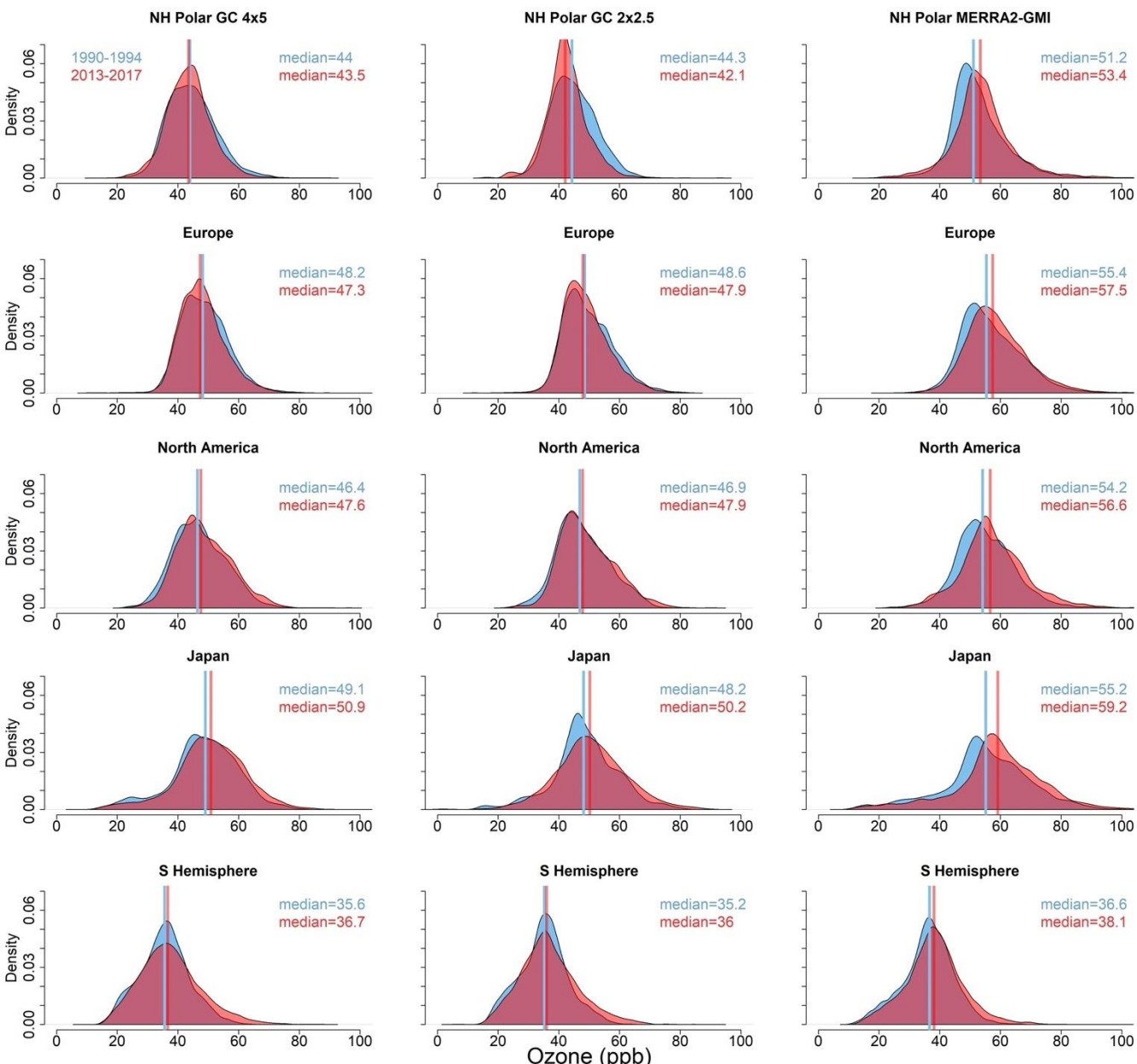

**Figure 8. Ozone distribution shifts from 1990-1994 (blue) and 2013-2017 (red) for all sites, broken into five regions in the GC 4x5,**
**GC 2x2.5, and MERRA2-GMI simulations. GC 4x5 refers to the GEOS-Chem v12.9.3 simulations at 4°x5°, GC 2x2.5 is the same**

model at 2°x2.5°, and MERRA2-GMI refers to the NASA GEOS GMI at ~50km resolution. Median concentrations are shown with vertical lines, and the corresponding values are recorded inset.

Increases in 5th percentile ozone across North America are marginally captured by all models, but only MERRA2-GMI captures
that trend over the NH Polar region and Europe. Shifts of the entire distribution that are observed over the Southern Hemisphere and Japan are captured by all models, although these shifts are typically underestimated (SH: 2 ppb observations, range of 1.1-2.8 ppb from models; Japan: 6.1 ppb observations, range of 1.8-4 ppb from models), with MERRA2-GMI replicating the shifts most reliably.

It is unlikely that the differences in trends between GEOS-Chem and MERRA2-GMI are primarily due to differences in the underlying emissions inventories. MERRA2-GMI used the MACCity inventory, and GEOS-Chem used the CEDS inventory. Typically, CEDS estimates higher magnitudes of $NO_x$ emissions and larger trends than MACCity (Fig. S7). However, we find that GEOS-Chem (using CEDS) produces smaller ozone trends than MERRA2-GMI, which suggests that the trend differences between models are more likely to be due to factors other than the emissions inventories, such as model resolution.

## 4.2 Model reproduction of ozone trends at the surface

Average trends in daytime ozone at surface locations overlap between models and observations (Fig. 9), although individual sites are typically not captured well. The average observed increasing surface ozone trend is $1.0 \pm 0.8$ ppb decade$^{-1}$, and all simulations overlap (GC 4x5: $0.6 \pm 0.8$ ppb decade$^{-1}$, GC 2x2.5: $0.6 \pm 0.7$ ppb decade$^{-1}$, MERRA2-GMI: $1.4 \pm 1.0$ ppb decade$^{-1}$).
$^{1}$). The direction of trends at the surface is generally captured by MERRA2-GMI, with the model capturing increasing trends at 67% of the surface sites also exhibiting increasing trends. Both GC simulations perform more poorly, with GC 2x2.5 capturing increasing trends at 37% of sites, and GC 4x5 capturing increasing trends at just 19% of sites. In both GC simulations, the trends predicted by the models at many locations, especially over North America and Europe, are opposite in sign to trends in the observations. At high elevation sites, which are more representative of regional air, the models do a better job of
predicting the observed direction, but do not capture the magnitude of trends. At these sites, MERRA2-GMI captures the sign of the trends at 5 of 8 sites, but underestimates these trends by 0.3 ppb decade$^{-1}$ on average. Both GC simulations capture the sign of the trends at 6 of the 8 high elevation sites, but the 4°x5° simulations overestimates the trends by 0.8 ppb decade$^{-1}$ on average and the 2°x2.5° simulation overestimates the trends by 0.5 ppb decade$^{-1}$ on average. The directions of regional changes are captured well by the models, but resolutions may be too coarse to get the surface trends at individual locations, especially
in the GC simulations.

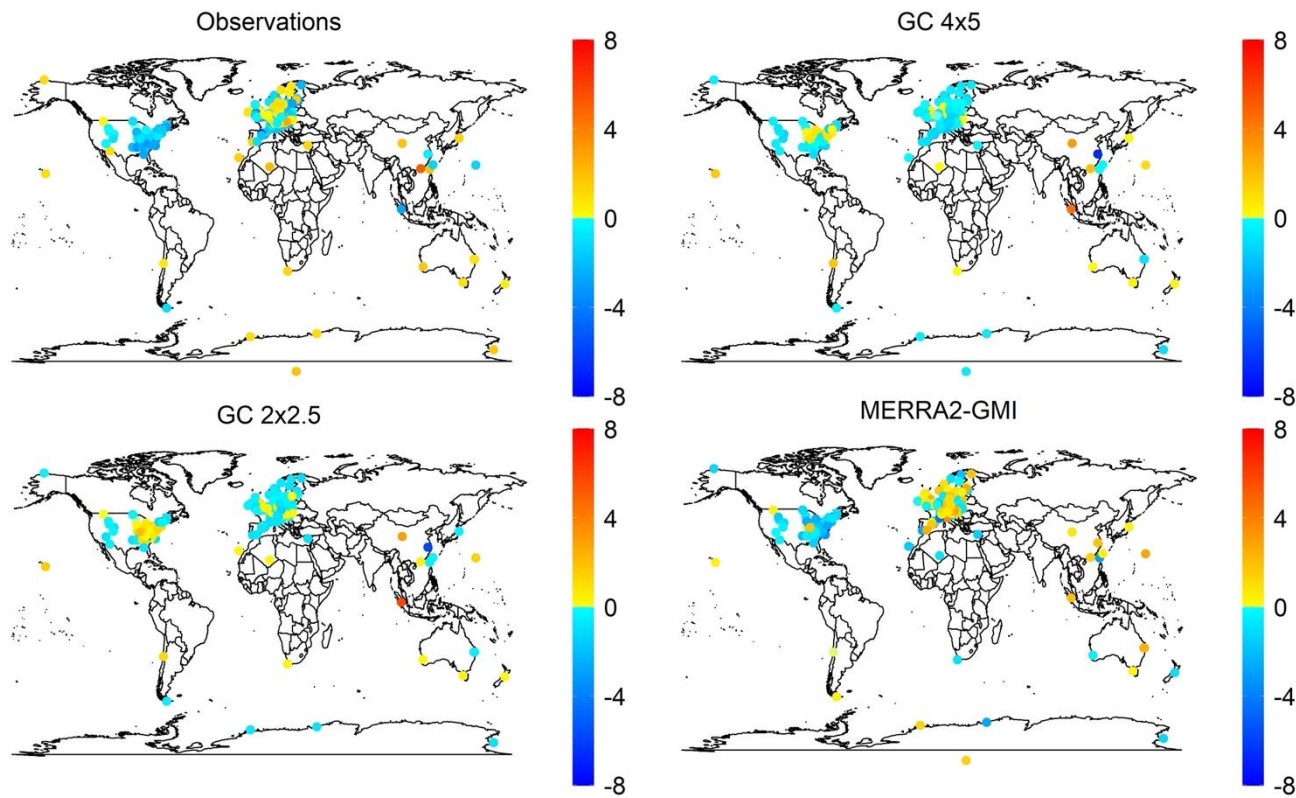

**Figure 9. Decadal trends (ppb decade⁻¹) at surface locations in TOAR-compiled observations, GC 4x5, GC 2x2.5, and MERRA2-GMI. GC 4x5 refers to the GEOS-Chem v12.9.3 simulations at the 4°x5° horizontal resolution, GC 2x2.5 is the same model at 2°x2.5°, and MERRA2-GMI refers to the NASA GEOS GMI at ~50km. Increases are shown in shades of red, and decreases are shown in shades of blue.**

Figure 10 shows shifts in surface regional ozone distribution medians in the models. All models qualitatively capture median shifts in Europe, North America, and the Southern Hemisphere, but models tend to underestimate these shifts. GC underestimates distribution shifts by 1.8 ppb on average, and MERRA2-GMI underestimates by 1.9 ppb on average. MERRA2-GMI reproduces the median shift in the Southern Hemisphere well (1.5 ppb in observations and 1.8 ppb in model). All simulations capture a median shift opposite in sign to the observations in Japan and the NH Polar region. The discrepancy between models and observations in both regions can be traced to the models' failure to capture the increased frequency of high-concentration ozone values during the 2010-2014 period. However, the models do capture the increase in frequency in low-concentration ozone values in Japan.

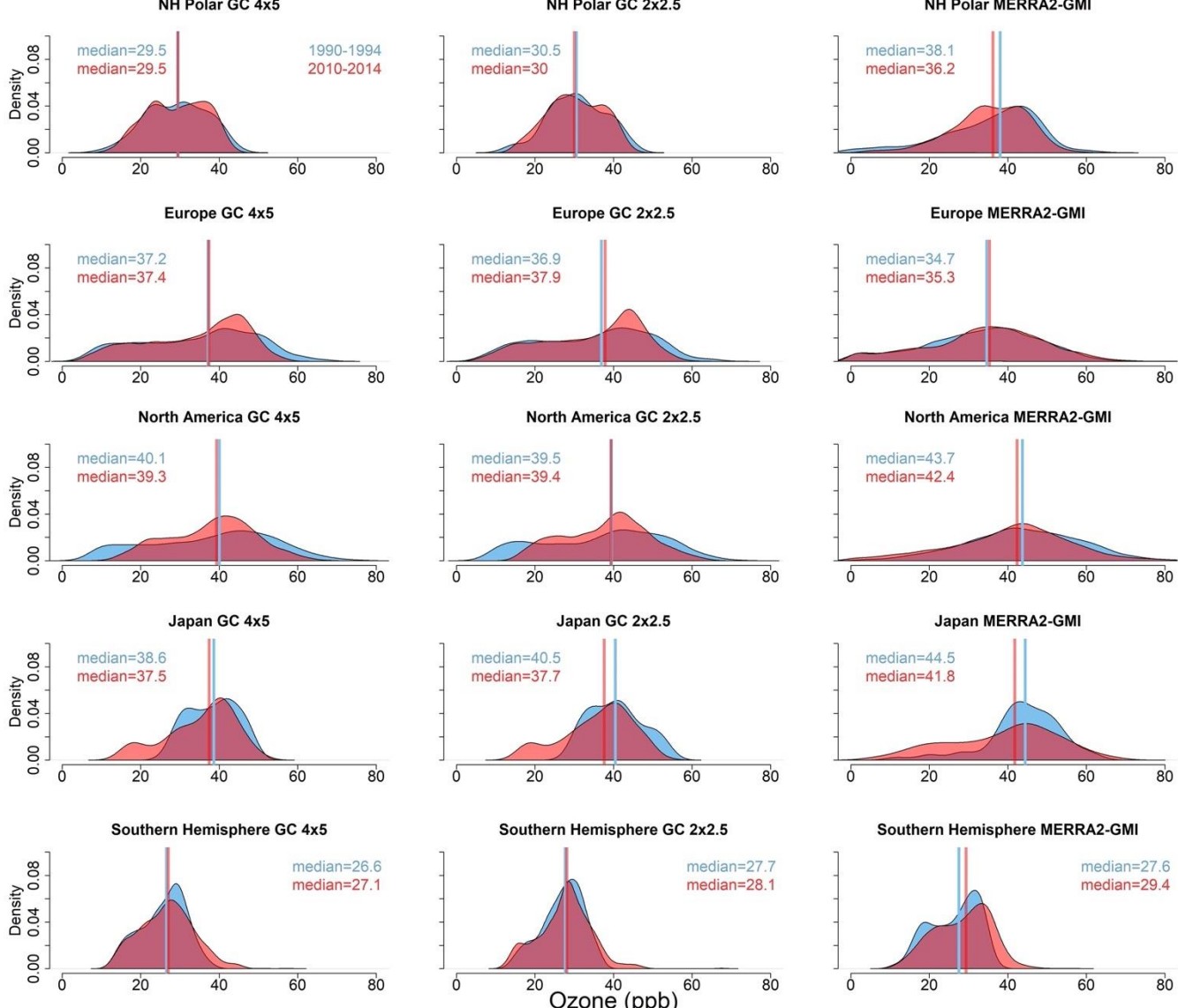

**Figure 10. Distribution shifts in ozone concentrations (ppb) between 1990-1994 (blue) and 2010-2014 (red) at surface sites, divided into five regions. GC 4x5 refers to the GEOS-Chem v12.9.3 simulations at the 4°x5° horizontal resolution, GC 2x2.5 is the same model at 2°x2.5°, and MERRA2-GMI refers to the NASA GEOS GMI at ~50km. Median concentrations are denoted with vertical lines, and the corresponding values are recorded inset.**

As explored earlier, observations suggest that increases in surface ozone are at least partially attributable to an increase in low-percentile ozone over North America, NH Polar, and Europe (Fig. 4). At the surface, increases in low quantile ozone values are captured by both GC simulations over North America, the NH Polar region, and Europe. Both GC simulations also capture the decreasing high tails in North America and Europe. In contrast, MERRA2-GMI does not capture the observed increases of

low quantile ozone at the surface in North America or Europe, and it does not capture the decreasing high tails in Europe. While all models capture the increasing high tail in Southern Hemisphere observations, the increase in frequency of low-concentration ozone values is reproduced only by GC 2x2.5.

## 4.3 Low model ozone burden in recent version of GEOS-Chem

While models tend to underestimate ozone increases globally, we find that the model ozone burdens in GC and MERRA2-GMI show global increases throughout the timeframe (Table 3), suggesting that the models capture at least some portion of the global ozone increase from 1980-2017. However, each of our simulations shows a smaller ozone burden than previous

analyses and model intercomparisons (Table 4). MERRA2-GMI gives an overall ozone burden that is ~10% lower than other estimates on average. In GC simulations, the magnitude of the ozone burden is considerably lower (by ~14-18%) than in a previous version (GEOS-Chem v10-01) and other model intercomparisons. Table 4 also summarizes chemical production, chemical loss, and dry deposition terms, and these are all lower in GC than in most other models. The only term in the ozone budget to increase between model versions is STE, which increases from the earlier version by 161 Tg/yr on average and

places it in the range of other models.

Systemically low model ozone burdens, especially in the northern mid-latitude free troposphere, are a known issue in recent versions of GEOS-Chem (Mao et al., 2021; Murray et al., 2021). We find that the underprediction of free tropospheric ozone persists across the last 4 decades of simulations, particularly in winter-and-springtime middle-to-high latitudes. While surface

ozone tends to be overestimated by GC (as well as MERRA2-GMI), FT ozone in GC is underestimated by ~10 ppb (Fig. S8). These underestimates may be caused by recent model developments such as improved halogen chemistry (Wang et al., 2021) or $NO_y$ reactive uptake by clouds (Holmes et al., 2019) that have increased sinks of ozone or $NO_x$. Neglect of lightning-produced oxidants may also be responsible for the ozone underestimates (Mao et al., 2021). Shah et al. (2022) found that including particulate nitrate photolysis in a recent version of GEOS-Chem increases ozone concentrations by up to 5 ppb in

the northern extratropics FT, which is not yet included in the model but will help to resolve this discrepancy in future analyses. By comparison, MERRA2-GMI and the earlier version of GEOS-Chem, both without the above model updates, nearly ubiquitously show ozone values that are much higher and closer to observations, and values are within 5% of observations at northern mid-latitudes in both simulations, although MERRA2-GMI tends to overestimate FT ozone at mid- and high-latitudes (Fig. S8) (Hu et al., 2017). However, it is important to note that the earlier version of GEOS-Chem does not perform better

than the more recent version in capturing long-term trends (Fig. S9), as it yields less than 10% of observed trends from 1990-2010. Such widespread model underestimation of tropospheric ozone across a long period highlights the need for better understanding of the processes that promote ozone production, such as VOC chemistry, biomass burning emissions, or the chemical evolution of smoke plumes (Bourgeois et al., 2021; von Schneidemesser et al., 2016). Improvements are especially important in the FT, where long-term transport of ozone is critical to understanding tropospheric ozone trends.

**Table 3. Ozone burden in 1980 and 2017, recorded in Tg O3.**

| | GC12 4x5 | | GC12 2x2.5 | | MERRA2-GMI | |
|---|---|---|---|---|---|---|
| | **1980** | **2017** | **1980** | **2017** | **1980** | **2017** |
| **Ozone Burden (Tg O₃)** | 280 | 313 | 272 | 301 | 300 | 323 |

**Table 4. Ozone budget terms in various model studies, with target years of simulations identified in the first column of the table. The standard deviations describe the spread among models in the model intercomparisons.**

| Model or model intercomparison | Sources (Tg/yr) | | Sinks (Tg/yr) | | Burden (Tg) |
|---|---|---|---|---|---|
| | **Chem Prod** | **STE** | **Chem Loss** | **Dry Dep** | |
| **GC10[a] (2012-2013)** | 4960 | 325 | 4360 | 910 | 351 |
| **ACCMIP[b] (2000)** | 4880 ± 850 | 480 ± 100 | 4260 ± 650 | 1090 ± 260 | 337 ± 23 |
| **IPCC AR6 (1995-2004) (CMIP6)[c]** | 5283 ± 1798 | 626 ± 781 | 4108 ± 486 | 1075 ± 514 | 347 ± 30 |
| **IPCC AR6 (2005-2014) (CMIP6)[c]** | 5530 ± 1909 | 628 ± 804 | 4304 ± 535 | 1102 ± 538 | 356 ± 31 |
| **TOAR[d] (2000)** | 4937 ± 656 | 535 ± 161 | 4442 ± 570 | 996 ± 203 | 340 ± 34 |
| **GC12 4x5 (1980-2017) (This work)** | 4077 | 615 | 3741 | 818 | 299 |
| **GC12 2x2.5 (1980-2017) (This work)** | 4269 | 497 | 3802 | 805 | 289 |

[a]Hu et al. (2017)

[b]Young et al. (2013)

[c]CMIP6: Coupled Model Intercomparison Project Phase 6; Griffiths et al. (2020)

[d]TOAR: Tropospheric Ozone Assessment Report; Young et al. (2018)

## 5 Potential reasons for model trend underestimates

### 5.1 Previously identified issues

Previous analyses have identified significant challenges facing models in reproducing observed tropospheric ozone trends in recent decades (Parrish et al., 2014; Young et al., 2018). In the northern hemisphere mid-latitudes, chemistry-climate models were only able to reproduce ~50% of the observed ozone trend (Parrish et al., 2014), consistent with our current analysis using chemical transport models. Another analysis using GEOS-Chem from 1995-2017 found that the model underestimated global

ozone trends compared to aircraft measurements and that aircraft emissions are a potential source of trend underestimation in the model (Wang et al., 2022). Tarasick et al. (2019) also pointed out the role of data representativeness: uncertainty in estimated observational trends stems largely from data representativeness rather than the accuracy of historical data, pointing to the importance of increasing ozone monitoring station number and frequency, especially when the evaluation of model skill necessarily relies on comparison to sparse datasets. The models examined in this work capture the general tendency of

increasing ozone from 1980-2017, and the multi-model average increase in global tropospheric ozone burden is 10% or 28 Tg (Table 3). However, they often underestimate tropospheric ozone trends at globally distributed sites (60% of trend captured with MERRA2-GMI, <15% for GC). Our findings that models are not able to reproduce recent ozone trends contrast with an analysis of GEOS-Chem and GISS-E2.1 that found the model accurately reproduced preindustrial ozone concentrations (Yeung et al., 2019). Notably, the GEOS-Chem simulations in that analysis were performed by running the standard model

without anthropogenic combustion and fertilizer sources. This result implies that a large issue in reproducing recent decadal trends may come from uncertainties in anthropogenic emissions, including neglected precursor emissions (Granier et al., 2011; Hassler et al., 2016) and underestimated aircraft emissions (Wang et al., 2022). Although the Yeung et al. (2019) results imply that natural sources are well-represented in models, natural sources of $NO_x$ and VOCs such as lightning, biogenic emissions, and soils are subject to many uncertainties: this includes land surface properties, the impact of land use change on biogenic

VOC emissions and ozone dry deposition (Tai et al., 2013; Fu and Tai, 2015), meteorological variables, and the sensitivity of ozone chemistry to emissions (Young et al., 2018; Banerjee et al., 2014; Hudman et al., 2012).

Another possible source of uncertainty in reproducing ozone trends is model representation of STE, which plays an important role in driving interannual variability and helps explain ozone changes that are not attributable to emissions changes alone (Liu

et al., 2017, 2020; Ordóñez et al., 2007). Previous studies have suggested that STE has increased over the last few decades (Neu et al., 2014; Griffiths et al., 2020) and is projected to increase over the next century due to increasing greenhouse gas emissions that strengthen Brewer-Dobson circulation, enhancing mean advective transport (Butchart et al., 2006; Hegglin and

Shepherd, 2009; Abalos et al., 2019). This increase in STE has been found to contribute to increases in tropospheric ozone in regions including North America, China, and the Southern Hemisphere (Liu et al., 2020; Xu et al., 2018; Lu et al., 2019). Recent analyses using an earlier version of GEOS-Chem suggests that STE in models may not be sufficient at high northern latitudes (Hu et al., 2017; Jaeglé et al., 2017). An issue with CTM simulations is that they require the aggregation of meteorological fields from their native resolution both spatially and temporally, which can cause losses in transport, especially vertical transport (Yu et al., 2018). Of the models we evaluate, MERRA2-GMI most accurately captures trends from 800-400 hPa and at the surface, perhaps due to its finer resolution that allows the meteorological products to be used at native resolution (~50km). The coarse resolution of the GC simulations means that remote sites can exist in the same grid cell as urban areas, limiting accurate representation of ozone in areas with sharp gradients (Lin et al., 2017).

## 5.2 Sensitivity simulations

Sensitivity simulations can provide further evidence behind model issues in reproducing ozone trends. Using the GC 4x5 simulation, we perform two sensitivity tests to examine the impact of emissions and meteorology on ozone trends from 1980-2017: 1) constant anthropogenic emissions ('Meteorology') and 2) constant meteorology ('Emissions'). In the 'Meteorology' simulation, all changes in ozone concentrations result from changes in meteorology, as anthropogenic emissions are cycled annually at 1980 values. Note that, in the 'Meteorology' simulation, only anthropogenic emissions from the CEDS inventory are cycled (e.g., $NO_x$, $SO_2$, CO, $NH_3$, NMVOCs, black carbon, and organic carbon). Conversely, in the 'Emissions' simulation, ozone changes stem from changes in emissions, with the meteorology cycled annually at 1980 values. Additionally, we examine stratospheric influence on tropospheric ozone using MERRA2-GMI's STO3 tracer (described in detail in Sect. 2.2.2) (Liu et al., 2020). Comparison of STO3 trends and tropospheric ozone trends within the MERRA2-GMI model can reveal the extent to which model trends at a given location are driven by stratospheric ozone. This can be a substantial effect, and a previous analysis by Griffiths et. al (2020) found that an increase in STE drove a small increase in tropospheric ozone burden from 1990-2010.

Figure 11 shows that, in GEOS-Chem, ozone trends at different altitudes are driven by different processes. At higher altitudes (i.e., 400 hPa), dynamics are an important driver of base GC trends in Europe and the NH Polar region. Here, the 'Meteorology' simulation accounts for the majority of trends in the base simulation at 400 hPa, while the 'Emissions' simulation shows opposite trends to the base. This result suggests that changing meteorological fields and dynamics such as intra-hemispheric transport and vertical transport from the stratosphere drive the ozone changes in the base simulation over these regions. At 600 and 800 hPa, meteorological fields still play a role in driving base simulation ozone trends, but emissions play a larger role closer to the surface. Non-anthropogenic emissions (e.g., soil $NO_x$, lightning, or biogenic VOCs) are not held constant in the

'Meteorology' run, and some of the ozone trend contribution at lower altitudes in this simulation may also be attributed to

these natural emissions.

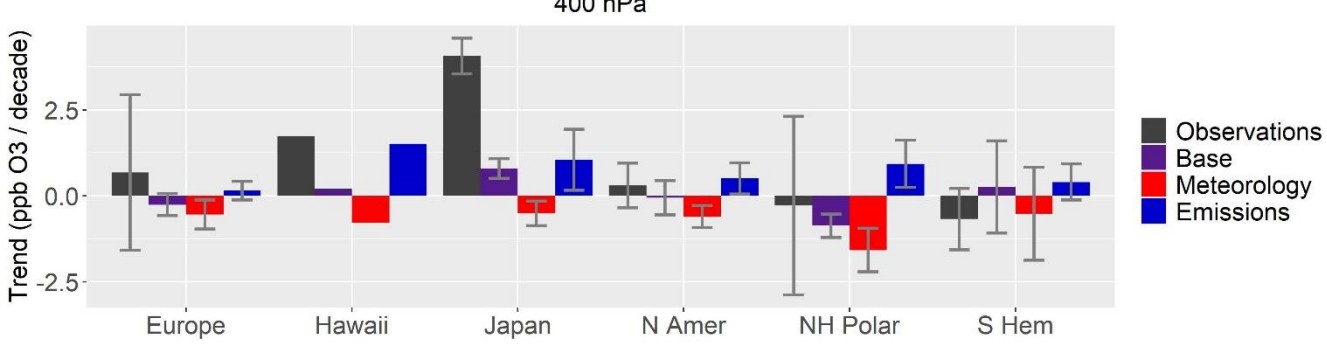

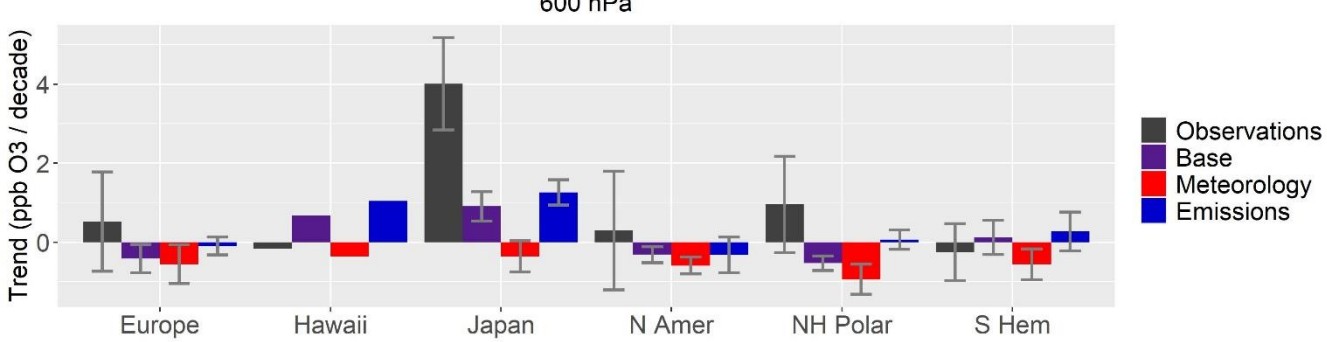

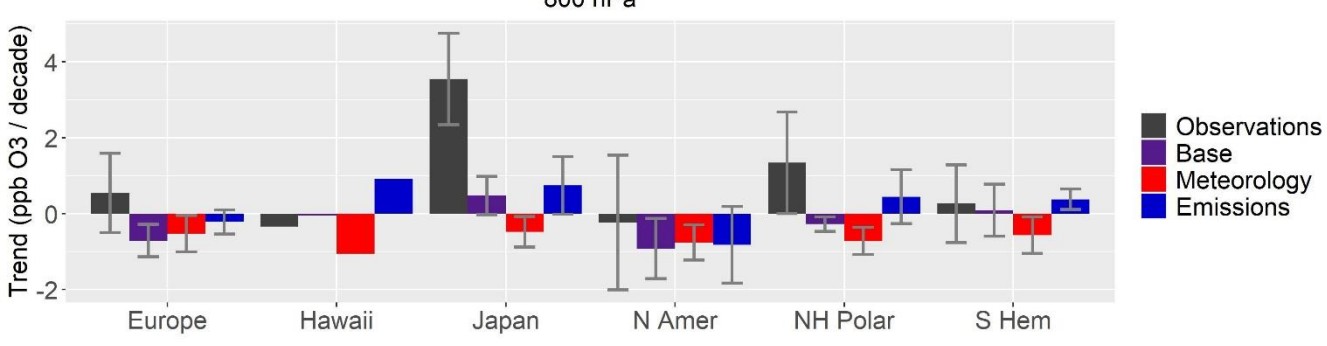

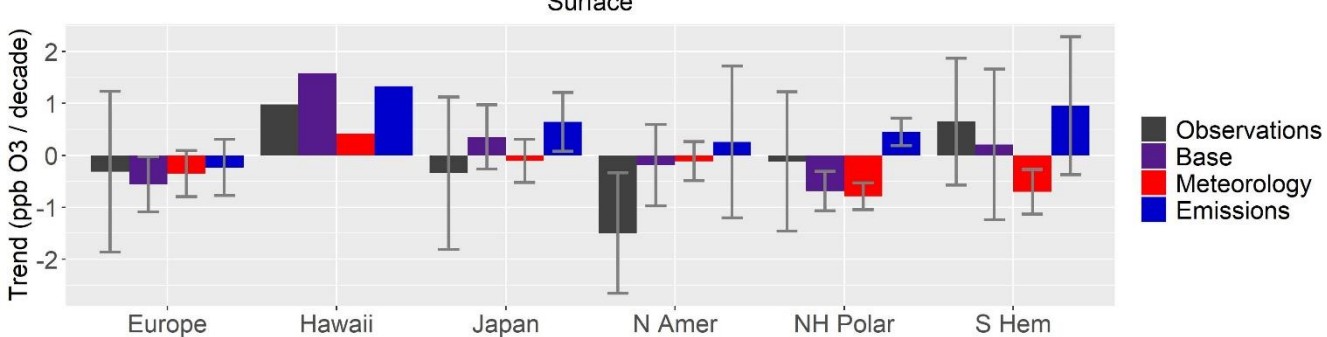

Figure 12, which investigates the role of STO3 in explaining ozone trends in MERRA2-GMI, also shows the importance of transport for understanding ozone trends. At 400 hPa over Europe, North America, and the NH Polar region, ozone trends are largely attributable to the stratospheric ozone influence. This aligns with the GEOS-Chem sensitivities that suggest meteorological inputs drive model trends at 400 hPa. Stratospheric influence is also prevalent at lower pressure levels for Europe and North America, consistent with previous analyses of ozone trends over these regions (Liu et al., 2020; Ordóñez et al., 2007). At the surface, the influence of STE is negligible in all regions. Importantly, MERRA2-GMI captures trends at 400 hPa remarkably well in Europe, North America, and the NH Polar region, which can be attributed to the ability of MERRA2-GMI to capture STE, likely due to its high resolution (Knowland et al., 2017). Model ability to capture vertical transport is important in reproducing ozone trends. GEOS-Chem and MERRA2-GMI show similar stratospheric trends (Fig. S10) but different trends at 400 hPa (Figs. 11 & 12), suggesting that transport from the stratosphere is most important for capturing trends at 400 hPa. Increases in MERRA2-GMI STO3 in the troposphere may stem from both changes in STE dynamics and recovery of the ozone hole. MERRA2-GMI has been shown to capture a decrease in lower stratospheric ozone in the northern extratropics from 1998-2016, when ozone-depleting substances were no longer increasing (Wargan et al., 2017). This decreasing trend was attributed to changes in lower stratospheric ozone circulation that may be due to climate change, but evidence for this is unclear. This decrease is offset by an increase in upper stratospheric ozone due to ozone layer recovery. The extent to which either dynamics or ozone recovery impacts increasing STO3 in MERRA2-GMI is currently difficult to quantify.

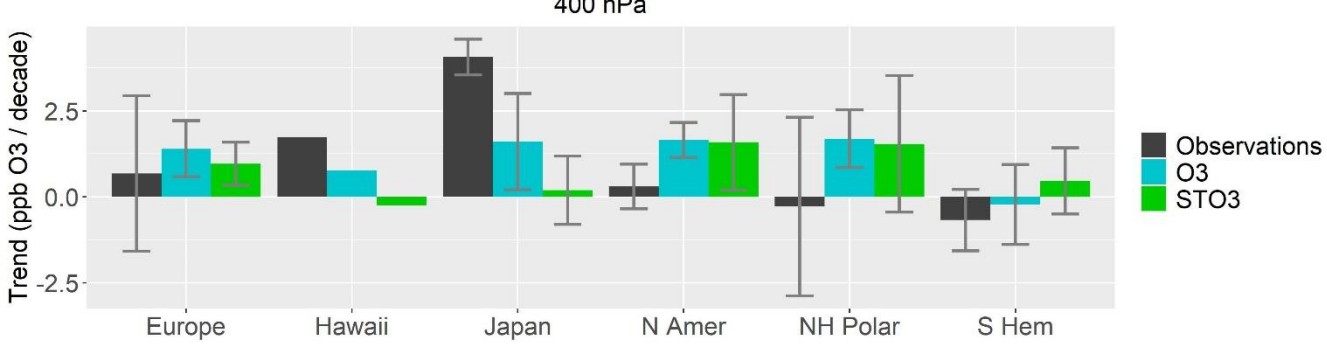

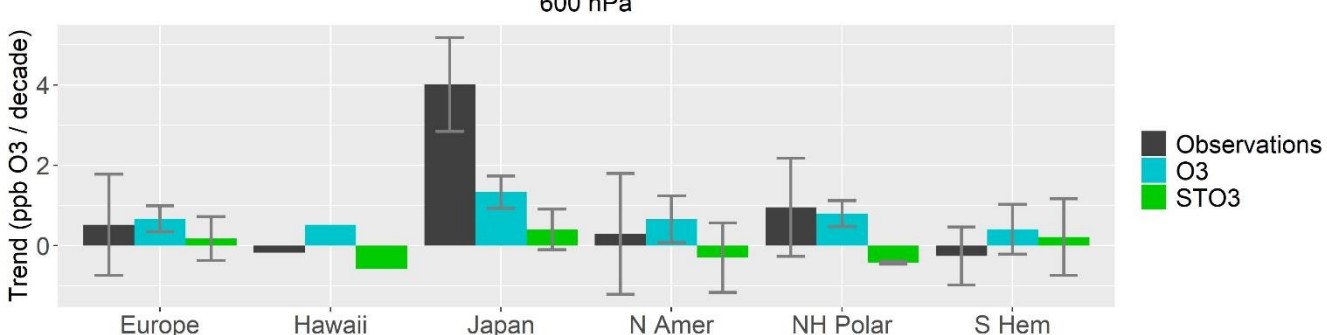

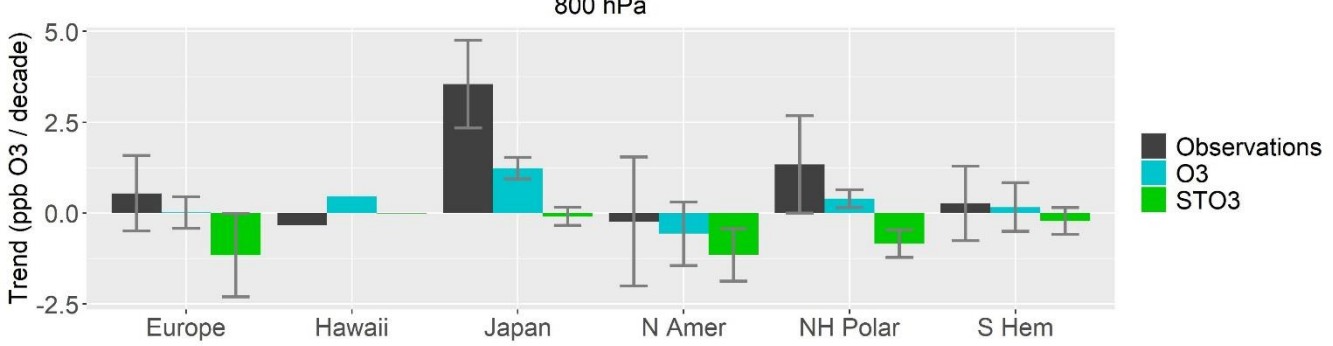

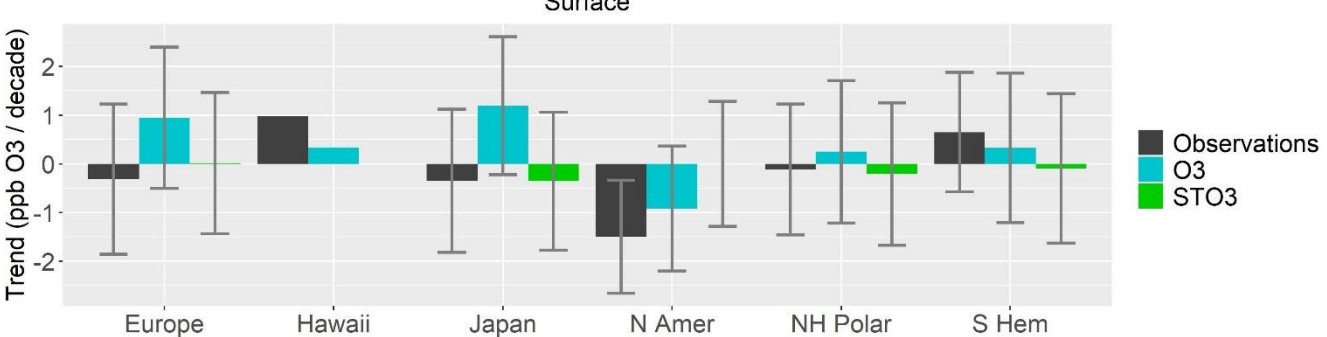

**Figure 12. Trends in tropospheric ozone in observations and in MERRA2-GMI at four pressure levels (surface, 800, 600, and 400 hPa) from 1990-2017, averaged over 6 regions. Observed ozone trends at ozonesonde and surface sites (black bars) are compared with MERRA2-GMI ozone (blue bars) and with STO3 (green bars), a tracer of the influence of stratospheric ozone in the troposphere (green bars). Gray thin bars denote the standard deviation across sites.**

In the other regions examined (Hawaii, Japan, and the Southern Hemisphere), the 'Emissions' simulation is able to explain more of the simulated ozone trend than the 'Meteorology' simulation. MERRA2-GMI agrees with GEOS-Chem in Japan and the Southern Hemisphere in that transport of ozone, either horizontally or from the stratosphere, does not explain ozone trends well at most pressure levels. This is in contrast with a recent analysis from Lu et al. (2019), which attributes observed Southern Hemisphere ozone changes primarily to changes in large-scale dynamics, although their focus was austral autumn. The large uncertainty bars in Figs. 11 and 12, which represent the standard deviation of trends across sites, show that the magnitudes of ozone trends and the primary drivers of these trends can vary across individual sites in a region. Future work must therefore focus on optimizing both emissions estimates and transport parameterizations in models to best capture observed ozone trends. Our model evaluations also reveal that the recent version of the GEOS-Chem model underpredicts free tropospheric ozone over the past 4 decades, particularly in the winter-and-springtime Northern extratropics. Such widespread model underestimation of tropospheric ozone highlights the need for better understanding of processes that promote model ozone production.

## 6 Conclusions

We have analyzed global ozone trends at 25 ozonesonde sites from 1990-2017, with 9 of those sites extending back to the 1980s. We show that ozonesondes launched at least 3 times per month are sufficient to capture tropospheric ozone trends. Across all sites in all regions, we find increases in tropospheric ozone from 800-400 hPa at 15 sites average $1.8 \pm 1.3$ ppb decade$^{-1}$ ($3.5\% \pm 2.6\%$ decade$^{-1}$), with relative trends slightly larger closer to the surface. Trends at high elevation sites, which sample air in the lower troposphere or free troposphere depending on location, closely match the trends we find from ozonesonde data, adding confidence to the ability of ozonesondes to robustly capture long-term trends in ozone. While most surface sites (62%) in the United States and Europe exhibit decreases in high ozone values due to regulatory efforts, 73% of global sites outside those regions (24 of 33 sites) show increases from 1990-2014. In all regions, increasing ozone trends both at the surface and aloft are at least partially attributable to increases in 5$^{th}$ percentile ozone, consistent with a potentially substantial impact of the largest sources of ozone precursor emissions shifting from the mid-latitudes toward the tropics. In the Southern Hemisphere and Japan, high quantile ozone also increases in response to changing emissions and dynamics.

Reproduction of ozone trends in models is essential to understanding ozone radiative forcing and the tropospheric ozone budget. We performed a model evaluation using three simulations comprising different emissions inventories, chemical schemes, and resolutions. To achieve the best model-measurement comparison of trends through the vertical column, we

sampled each model at the same time (within 3 hours) and location of each individual ozonesonde launch. Despite using the latest model updates and sampling as accurately as possible, models are not able to replicate long-term ozone trends throughout the troposphere, often underestimating the trend. MERRA2-GMI captures ~75% of the trend, while GEOS-Chem only captures

<20%. MERRA2-GMI performs better than GEOS-Chem in the northern mid-latitudes free troposphere, where it captures 44% of the trend, likely due to the higher resolution of this model. Similarly, daytime surface ozone trends are not reproduced well by GEOS-Chem, but MERRA2-GMI reproduces the direction of trends at 67% of sites. However, shifts in ozone percentile distributions from 1990-2017 are underestimated by all models. Even though models underestimate ozone increases and ozone burdens in GEOS-Chem are substantially lower than early versions and all other models, each model shows an

increase of ~10% in total ozone burden from 1980-2017, indicating that models capture at least some of the global tropospheric ozone increase over the past few decades. Sensitivity simulations suggest that, in the northern mid- and high-latitudes, dynamics such as STE are important for reproducing ozone trends in models in the middle and upper troposphere, while emissions are important closer to the surface. Our work thus points to the importance of constraining both emissions trends and transport processes in improving the modeled representation of global ozone trends.

**Data and code availability**

The data and R code used in this study are available from the authors upon request. Data from the MERRA2-GMI simulation is archived at https://acd-ext.gsfc.nasa.gov/Projects/GEOSCCM/MERRA2GMI/.

**Author contributions**

LH and LJM designed the research. AC performed GEOS-Chem v12.9.3 model simulations, data analysis, and wrote the paper.

LH performed the GEOS-Chem v10-01 model simulation. LDO performed the MERRA2-GMI model simulation, and JL assisted in retrieving data from that simulation.

**Competing interests**

The authors declare that they have no conflict of interest.

**Acknowledgements**

This research was supported by NOAA Climate Program Office's Atmospheric Chemistry, Carbon Cycle, and Climate program, grant number NA19OAR4310174 (Montana)/ NA19OAR4310176 (Harvard). The authors would like to acknowledge high-performance computing resources and support from Cheyenne (doi:10.5065/D6RX99HX), provided by the National Center for Atmospheric Research (NCAR) Computational and Information Systems Laboratory and sponsored by

the National Science Foundation, and the University of Montana's Griz Shared Computing Cluster (GSCC). The authors thank WOUDC for the public availability of ozonesonde data, which can be accessed at doi:10.14287/10000001. The authors also thank the National Oceanic and Atmospheric Administration's Global Monitoring Laboratory and the National Aeronautics and Space Administration's (NASA) Southern Hemisphere ADditional OZonesondes (SHADOZ) team (PI: Dr. Ryan M. Stauffer, Founding PI: Dr. Anne M. Thompson) for the public availability of their ozonesonde data. The authors also acknowledge the ongoing work toward ozonesonde data homogenization, including substantial efforts from SHADOZ and HEGIFTOM. The authors thank Forchungszentrum Julich for funding of the TOAR database development and its maintenance, as well as its data providers and Martin Schulz for providing publicly available compiled ozone data. The authors also thank the NASA MAP program and the NASA Center for Climate Simulation (NCCS) for supporting the MERRA2-GMI simulation.

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
