# Peer review of "Multidecadal increases in global tropospheric ozone derived from ozonesonde and surface site observations: Can models reproduce ozone trends?"

_Atmospheric Chemistry and Physics, 2022_

## Author Response (AR1)

We are grateful for the Reviewers and Community members who have taken the time to read and improve this manuscript. Our response to each comment is shown in blue. Where we include text changes, the original text is shown in blue, and the changed text is shown in red.

Community Comment:

Comment on "Multidecadal increases in global tropospheric ozone derived from ozonesonde and surface site observations: Can models reproduce ozone trends?" by A. Christiansen, et al.

First, we would like to express that this paper was well-motivated and written, has important concluding messages, and was a pleasure to read. This comment will cover two topics/points of concern:

- Suggested additional references and discussion
- Choice of data archives, station selection, and data version caveats

**Topic 1: References and Discussion**

- Tropical free-tropospheric (FT; above 5 km) and lower stratospheric ozonesonde trends from 1998-2019 were recently published in Thompson et al. (2021; https://agupubs.onlinelibrary.wiley.com/doi/full/10.1029/2021JD034691). These represent a definitive reference for the model comparisons here because they are based on reprocessed SHADOZ data. A good place to reference is near lines 69-81. On an annually-averaged basis, the FT ozone trends from SHADOZ are fairly weak at 5 zonally distributed tropical sites +(1-4)%/decade, but the seasonal variability in the ozone trends was significant at all sites, with trends up to +(10-15%)/decade in some months. Although only one station, Nairobi (also see second Topic), in your study overlaps with ours, we note that quantifying the seasonal variability of trends magnitude is important for diagnosing the trends and potential dynamic factors. At Nairobi, for example, FT ozone increases 5-10%/decade in February-April, but decreases in the mid-FT August-September (Fig 6 in Thompson et al.). We also observed that the FT ozone increase is strongest during the seasonal minimum (e.g., analogous to your $5^{th}$ percentile positive ozone trends). Trends output from the Thompson et al., (2021) paper is located here so you can make direct comparisons: https://tropo.gsfc.nasa.gov/shadoz/SHADOZ_PubsList.html.

Thank you for bringing these trends to our attention. We have added trend information to our introduction to include the reprocessed SHADOZ data. We mention these SHADOZ trends in Lines 83-87, which reads:

"Ozone measurements from the Southern Hemisphere Additional Ozonesondes (SHADOZ) show increasing FT ozone in some parts of the tropics, with tropical South American and Asian sites showing annual average increases of 5% decade$^{-1}$ from 1998-2019 (Thompson et al., 2021). However, large regions of the tropics do not show annual increases, with increasing ozone limited to certain seasons at most stations (e.g., Nairobi FT ozone increases 5-10% decade$^{-1}$ during February-April but does not increase on an annual basis) (Thompson et al., 2021)."

We discuss the seasonality of trends further in Section 3.2. We specifically discuss Nairobi trends here, as that is the one site that overlaps between our study and Thompson et al. This section now reads:

"It is important to note that we have not performed a seasonal analysis of ozonesonde data. Analyses of ozonesonde sites in the tropics point to the seasonal variability of ozone and show that trends are driven primarily by changes during certain months. For example, Thompson et al. (2021) did not find significant trends at Nairobi, Kenya, on an annual basis (consistent with our results), but found that FT ozone increased during February-April by 5-10% decade$^{-1}$ while decreasing during August-September. Other tropical sites show similar patterns – annual trends are insignificant, while seasonal trends are much larger. A seasonal analysis is beyond the scope of this paper, as the 3-4 launches per month may not give enough information for robust monthly or seasonal trend analysis. Future investigations of ozone trends should consider the impact of specific months or seasons, provided it can be done in a statistically meaningful way, to aid in identifying drivers of trends."

- The MERRA-2 GMI simulation used in this study was extensively evaluated against ozonesonde data in Stauffer et al., (2019; https://agupubs.onlinelibrary.wiley.com/doi/full/10.1029/2019JD030257) from 1980-2016. In short, M2 GMI overestimates FT ozone in mid- and high-latitudes, underestimates FT ozone in subtropical and tropical latitudes, and has a notable step change in ozone after ~1998 associated with an observing system upgrade incorporated into MERRA-2. This step change in M2 GMI ozone may mostly lie above the altitudes that you examine and is perhaps not a concern (see Figure 2a in that paper). A high bias in M2 GMI surface ozone occurs across all latitudes compared to the ozonesonde measurements. These are important simulation characteristics to be aware of when discussing trends comparisons with real data, and this information would complement your Figure S4, for example.

Thank you for bringing this paper to our attention. We have noticed many of the characteristics you mention, and they can be seen in our Figure S7. We did not discuss these ozone over- and underestimates since our focus in this paper is on trend reproduction rather than magnitudes. However, we have added a statement regarding the simulation characteristics and the step change in Section 4.3. It reads:

"While surface ozone tends to be overestimated by GC (as well as MERRA2-GMI), FT ozone in GC is underestimated by ~10 ppb (Fig. S7). These underestimates may be caused by recent model developments such as improved halogen chemistry (Wang et al., 2021) or NO$_y$ reactive uptake by clouds (Holmes et al., 2019) that have increased sinks of ozone or NO$_x$. Neglect of lightning-produced oxidants may also be responsible for the ozone underestimates (Mao et al., 2021). Shah et al. (2022) found that including particulate nitrate photolysis in a recent version of GEOS-Chem increases ozone concentrations by up to 5 ppb in the northern extratropics FT, which is not yet included in the model but will help to resolve model-observation discrepancies in future analyses. By comparison, MERRA2-GMI and the earlier version of GEOS-Chem, both

without the above model updates, nearly ubiquitously show ozone values that are much higher and closer to observations, and values are within 5% of observations at northern mid-latitudes in both simulations, although MERRA2-GMI tends to overestimate FT ozone at mid- and high-latitudes (Fig. S7) (Hu et al., 2017)."

We do not anticipate that this step change in MERRA2-GMI will substantially impact our findings. This step change occurred from 6-12 km, which mostly lies above our analysis range. It is possible that pressures less than ~450 hPa may be impacted by this change, and we add a short discussion of this to Section 4.1. It reads:

"Notably, MERRA2-GMI typically performs better in the upper free troposphere than the GC simulations in the northern mid-latitudes, matching 44% of the observed trend from 600 to 450 hPa, while the GC simulations only capture 24%. An important note is that a significant step change occurred in MERRA2-GMI ozone after 1998 associated with an observing system upgrade incorporated into MERRA-2 (Stauffer et al. 2019). This step change impacts pressure levels mostly above our analysis range, and model ozone at pressures <450 hPa may be affected. MERRA2-GMI trends at pressures <450 hPa should thus be interpreted with caution."

**Topic 2: Ozonesonde Data Sources and Versioning**

- We were surprised to see a lack of tropical ozonesonde data included in this study, and wondered whether it was because only data from WOUDC and NOAA GML archives were sought. Especially for tropical ozonesonde data, WOUDC is not kept as up to date as SHADOZ (https://tropo.gsfc.nasa.gov/shadoz/Archive.html) and NDACC (https://www-air.larc.nasa.gov/missions/ndacc/data.html). Stations such as La Reunion, Pago Pago American Samoa, Paramaribo, etc. would probably qualify for inclusion based on your criteria if data were retrieved from SHADOZ instead of WOUDC.

We appreciate you bringing this to our attention. We agree that it would have been great to have been able to include more tropical ozonesonde data. Unfortunately, none of the sites in the SHADOZ archive (outside of Hilo and Nairobi, which we already include and for which we now use the homogenized data) had enough data to be considered in our analysis. Typically, these sites did not have enough observations per month consistently throughout the timeframe (>= 3 launches per month for at least 8 months of the year, with each season requiring 2 valid months), which excluded them from analysis according to our criteria. Since we are examining trends back to 1980, this requirement can be difficult to fulfill, but analyses investigating shorter timeframes should make use of this important collection of data. We add a statement in our methods (Section 2.1) to address why we did not use more tropical data:

"Updated tropical ozonesonde information is available from the Southern Hemisphere ADditional OZonesondes (https://tropo.gsfc.nasa.gov/shadoz/), but we did not include these data in this analysis because they did not meet our data requirements described below, typically due to not having enough profiles per month consistently throughout our timeframe."

- The global ozonesonde community is involved in an ongoing data reprocessing and homogenization effort (e.g., Tarasick et al., 2016; Van Malderen et al., 2016; Witte et al., 2017; Sterling et al., 2018; Witte et al., 2019; Ancellet et al., 2022). Homogenization accounts for changes in ozonesonde preparation and station procedures and reduces measurement biases associated with them. See, for example, our Wallops Island (NASA site back to the 1960s; Witte et al., 2019; https://agupubs.onlinelibrary.wiley.com/doi/full/10.1029/2018JD030098) station where "before" and "after" reprocessing comparisons are provided. The homogenized data versions are more accurate than non-homogenized data. See the preprint of Stauffer et al., (2022) at https://www.essoar.org/doi/abs/10.1002/essoar.10511590.1 for a summary of global ozonesonde network data quality, data sources, homogenized data availability, and links to the references above. The data at WOUDC for most stations are likely not homogenized. We know they are not for nearly all SHADOZ stations, for example. Caution must be exercised when calculating trends from non-homogenized data, and the time series should be evaluated for step-changes if non-homogenized data are used.

Thank you for bringing this to our attention. We have located 13 homogenized datasets out of our 25 sites from HEGIFTOM, SHADOZ, and NOAA and have compared trends from the original and homogenized datasets. Here, in Figure S4 shown below, we have analyzed the original and available homogenized data across consistent timeframes. Only one site, Eureka, shows a complete flip in the magnitude of the trend. Another site, Payerne, flips from insignificant trends to negative trends. While we find differences, sometimes substantial, in trend magnitude between these datasets, our overall conclusions have not changed. At most locations, ozone has been increasing since the 1990s. We include the homogenized data where possible, update all numbers and figures to reflect this, and add information to the methods section (Section 2.1) regarding at which sites homogenized data was used. We note that we are unable to use the homogenized dataset for Payerne in our analysis, as it has only been homogenized since 2002, a timeframe too short for our analysis. We include Figure S4 in the SI to illustrate the differences that homogenization makes, as well as to show that using this data does not change our overall conclusions. We also include Table 1, produced in response to Reviewer 1's comments, to aid readers in the identification of data sources, data ranges, and whether or not homogenized data was used. For the remaining 12 sites where homogenized data is not available, we ensure that it does not contain step changes, which is shown in Figure S1. We include Figure S2 for the homogenized sites as well as a point of comparison. The added text in Section 2.1 reads:

"Ozonesonde vertical profile measurements from 1980-2017 were downloaded from the World Ozone and Ultraviolet Data Center (WOUDC) (https://woudc.org/data/explore.php), the National Oceanic and Atmospheric Administration (NOAA) (ftp://ftp.cmdl.noaa.gov/ozww/Ozonesonde/), and the Harmonization and Evaluation of Ground-based Instruments for Free Tropospheric Ozone Measurements (HEGIFTOM) working group of the Tropospheric Ozone Assessment Report, Phase II (TOAR-II) (https://hegiftom.meteo.be/datasets/ozonesondes). The global ozonesonde community is currently reprocessing and homogenizing data to account for changes in ozonesonde preparation and procedures, with the goal to reduce measurement biases associated with these changes (Tarasick et al., 2016; Van Malderen et al., 2016; Witte et al., 2017; Sterling et al., 2018; Witte et

al., 2019; Ancellet et al., 2022). Where possible (12 of 25 sites), homogenized ozone profiles were used to ensure the most accurate ozone trends. Table S1 describes the ozonesonde profile information, dates, and whether the data is homogenized. While Payerne (Europe) has homogenized data, we use the original data since the site has only been homogenized since 2002. For data that are not yet homogenized, we check for evidence of step changes (Figs. S1 & S2). Updated tropical ozonesonde information is available from the Southern Hemisphere Additional OZonesondes (SHADOZ) (https://tropo.gsfc.nasa.gov/shadoz/), but we did not include these data in this analysis because it did not meet our data requirements described below, typically due to not having enough profiles per month consistently throughout our timeframe."

We also add an additional caution for interpreting the results of the mixed homogenized and non-homogenized data in Section 3.2:

"We caution that these results are derived from both homogenized and non-homogenized data depending on availability (see Table 1 for a list of homogenized sites). The impact of homogenization is shown in Figure S4, with homogenization affecting trend magnitudes but rarely the sign of the trends compared to non-homogenized data. Future analyses should use homogenized data as it becomes available to discuss quantitative ozone trends most accurately."

Our additional figures and tables are shown below:

[Figure]

**Figure S4. Differences in trends between homogenized (red) and non-homogenized ("Original", black) data. The impact is to modify the magnitude of trends, but the sign of the trends stays the same at most locations.**

[Figure]

**Figure S1. Annual average ozone profiles for ozonesonde data from the 12 non-homogenized sites from 1990-2017. No step changes are apparent in the data.**

[Figure]

**Figure S2. Annual average ozone profiles for ozonesonde data from the 13 homogenized sites from 1990-2017. No step changes are apparent in the data.**

- A note regarding the Stauffer et al., (2020) study referenced in your paper: The ozonesonde "dropoff" was found not to be a concern below about the 50 hPa level at affected stations, so the inclusion of all tropospheric data at "affected" stations should not harm your results, with the possible exception of Hilo, and, if added, Costa Rica. The Stauffer et al., (2022) study updates the status of this low bias problem, which still appears to be confined to stratospheric ozonesonde data at select stations.

Thank you for the correction on this study. We have updated the methods section (Section 2.1) to clarify this. It now reads:

"A recent study showed a drop in total column and stratospheric ozone measured by ECC instruments compared to satellite observations in the latter parts of their records for reasons still under investigation (Stauffer et al., 2020). We find that 5 of our 25 sites were impacted by these ozone measurement drops, although these drop-offs were typically limited to pressures above ~50 hPa, so our results should not be affected. Out of an abundance of caution, at these impacted sites, we used only data from before the unexplained sharp drop-off in ozone concentrations, as data before these drops is still considered highly reliable (Stauffer et al., 2020), and this resulted in the removal of up to one year of data at each affected site."

The manuscript presents an important piece of work to evaluate tropospheric ozone trends for 30 years or longer from ozonesonde and surface measurement and from two chemical models. The authors have conducted a careful analysis on the trend calculation, compared the consistency and inconsistency with previous studies, uncovered the model disability to reproduce the ozone trends, and discussed the possible reasons. In general this work is well-motivated, the analyses are comprehensive and easy to follow. Using two global models for long-term ozone trend analyses is particularly appreciable. Interestingly at almost the same time another paper comes up in ACPD using the same model (GEOS-Chem) to evaluate and attribute global tropospheric ozone trends (Wang et al., 2022, ACPD, in review), and I am glad to see that the two papers review the same important topic, and consistently point to the positive ozone trends from observations and the challenge for model to reproduce these trends.

I recommend this work to be published in ACP after moderate revisions. Below are some questions and comments that may help the authors to further strengthen their analyses and improve the presentation.

**1. Terminology.** The authors may use term "background ozone" or "baseline ozone" in literature review and analyses (e.g. Line 97, Line 474, Line 202, …). Please try to clarify the terms when they first appear in the text. The authors may have known that policy-relevant background ozone has a more rigorous definition in US as ozone concentration without national anthropogenic sources. This should not be the same as "rural" defined in TOAR. Is there a clear definition on "baseline ozone"?

Thank you for bringing this to our attention. We have revised our language for clarification. We have changed all instances to specify low-percentile (e.g., $5^{th}$, $10^{th}$) ozone rather than background. Where we are referring to policy-relevant background ozone (Line 142), we define it. In Lines 551-552, we specify that "baseline ozone" refers to ozone that is not influenced by local emissions and remain consistent throughout with using "baseline ozone" rather than background when we are referring to this definition.

**2. Trends from ozonesonde and their representativeness of free tropospheric ozone.** The robustness of trends derived from ozonesonde is an important topic being argued a lot. The authors have tried to make sure that they select ozonesonde sites with frequent sampling and the trends are consistent with surface sites nearby. I appreciate all the efforts. There are also studies suggesting that more profiles per month are required for robust trend quantification at a single ozondesonde site, and aircraft observations from the IAGOS project may provide better quantification of tropospheric ozone trends (Gaudel et al., 2020; Chang et al. 2020, 2021). I wonder whether the authors can try to reconcile their analyses with these existing literatures. It would be great if the authors can also utilize the aircraft (IAGOS) observations for evaluating the ozonesonde trends but if not some discussions are also acceptable.

In Section 3.2, we compared our free tropospheric ozone trends to those presented in Gaudel et al. (2020) and found very good agreement. We do not include our own analysis of IAGOS data since this dataset has already been extensively explored over our timeframe, but we add a few

sentences comparing these existing results to our ozonesonde findings. A further discussion of aircraft results is now included in Section 3.2:

"Free tropospheric and tropospheric column ozone measured by IAGOS also suggests that ozone has increased across the Northern Hemisphere since the 1990s (Gaudel et al., 2020; Petzold et al., 2015; Cohen et al., 2018) at an average rate of 2 ppb decade$^{-1}$, which agrees with the average $2.0 \pm 1.3$ ppb decade$^{-1}$ increase in Northern Hemisphere FT ozonesonde measurements. Although some variation is expected when comparing regions to individual sonde launch locations, our results show good agreement with previous analyses of FT ozone (700-300 hPa) since the 1990s using IAGOS flight data. Over Europe, Gaudel et al. (2020) found an increasing trend of $1.3 \pm 0.2$ ppb decade$^{-1}$, slightly lower than our result of $1.9 \pm 1.1$ ppb decade$^{-1}$ but within uncertainty. Gaudel et al. (2020) report an increase of $1.3 \pm 0.9$ over the Southeast US FT, which aligns with our findings in the upper troposphere at Wallops Island (Virginia, US; $0.8 \pm 0.3$ ppb/dec). Over Eastern North America, an increase of $1.7 \pm 0.4$ ppb decade$^{-1}$ is in good agreement with ozonesonde measurements at Goose Bay (Eastern Canada; $2.0 \pm 0.7$ ppb/dec). This remarkable agreement between ozonesondes and other measurement platforms lends further evidence that ozonesondes launching 3 times per month are able to capture long-term trends in tropospheric ozone."

We were not aware of the Chang papers and have included discussion of them in Section 3.2. The text is included below. We thank the Reviewer for bringing this to our attention.

"There is much discussion about the number of profiles needed for statistical analyses of global ozone trends, and recent studies have suggested that 14 profiles per month are needed (Chang et al., 2020). However, this number of profiles is not possible under the current ozonesonde sampling landscape. Here, we show that careful selection and treatment of ozonesonde data can lend important insights to global ozone trends that are highly vertically resolved. We note that these trends may not be considered globally representative, but rather they offer an additional insight into ozone changes over the past 30 years. That we find good agreement between ozonesonde trends and trends from other data sources suggests that ozonesonde information is an important part of the ozone monitoring landscape in determining global trends."

**3. GEOS-Chem sensitivity simulation.** I appreciate that the authors applied the UCX scheme in their GEOS-Chem simulation for a better presentation of stratospheric chemistry. I wonder when the authors fixed anthropogenic emissions in their sensitivity simulation, did ozone-depletion-species also be fixed in the model? Or in other words, whether stratospheric influences from GEOS-Chem should be analyzed from the "Meteorology" simulation or in the "Emission" simulation? This should be clarified both in Section 2.2.1 and in Section 5.2.

In our sensitivity simulation, we only held anthropogenic emissions from the CEDS inventory constant (e.g., NOx, SO2, CO, NH3, NMVOCs, black carbon, and organic carbon), thus ozone-depletion-species such as halogens were allowed to evolve over time. This means that the stratospheric influences are the same in the Base simulation and the Constant Emissions simulation. Differences in the Base and Constant Emissions simulations are most attributable to

changes in tropospheric chemistry arising from anthropogenic emissions near the surface. We have clarified this in Sections 2.2.1 and 5.2:

Section 2.2.1: "We performed two sensitivity tests at the coarse (4°x5°) resolution due to computational constraints. One simulation held anthropogenic emissions constant throughout 1980-2017. Note that only anthropogenic emissions in the CEDS inventory are held constant (e.g., $NO_x$, $SO_2$, CO, $NH_3$, NMVOCs, black carbon, and organic carbon). The other simulation held the meteorological condition as 1980 with varying anthropogenic emissions."

Section 5.2: "In the 'Meteorology' simulation, all changes in ozone concentrations result from changes in meteorology, as anthropogenic emissions are cycled annually at 1980 values. Note that, in the 'Meteorology' simulation, only anthropogenic emissions from the CEDS inventory are cycled (e.g., $NO_x$, $SO_2$, CO, $NH_3$, NMVOCs, black carbon, and organic carbon)."

**4. GMI vs GEOS-Chem simulations.** The authors use two chemical models and find that the GMI model produces larger ozone trends than GC. I am curious that whether the difference in anthropogenic emission inventory (and trends) may contribute. Could the authors show the emission trends used in the two model?

The emissions trends for the full 1980-2017 time period for MERRA2-GMI are not published in their data archive, so we use information directly from the emissions inventory, which only spans 2000-2010. Global NOx trends in MACCity and CEDS are plotted below, and both inventories show generally increasing global emissions trends. This plot shows that CEDS magnitudes and trends are higher than MACCity; however, MERRA2-GMI (which uses MACCity) predicts higher ozone than GEOS-Chem (which uses CEDS). Much of the differences between these inventories occur over areas with less monitoring information (e.g., NOx emissions between inventories are similar over well-monitored areas such as the United States and Europe). CEDS shows a sharper increase than MACCity in NOx emissions from 2000-2010 (0.8 Tg/yr in CEDS; 0.2 Tg/yr in MACCity). However, we see that MERRA2-GMI (which uses MACCity) is better at reproducing increasing trends than GEOS-Chem (CEDS), and it typically captures larger ozone trends. While the differences in emissions trends could contribute to the differences in ozone trends between the models, it is more likely that the differences are due to a combination of factors such as model resolution, chemistry, etc. We add a brief discussion in Section 4.1 and add the figure to the SI (Figure S7):

"It is unlikely that the differences in trends between GEOS-Chem and MERRA2-GMI are primarily due to the underlying emissions inventories. MERRA2-GMI used the MACCity inventory, and GEOS-Chem used the CEDS inventory. Typically, CEDS estimates higher magnitudes of $NO_x$ emissions and larger trends than MACCity (Fig. S7). However, we find that GEOS-Chem (using CEDS) produces smaller ozone trends than MERRA2-GMI, which suggests that the trend differences between models are more likely to be due to factors other than the emissions inventories, such as model resolution."

[Figure]

**Figure S7. Trends and magnitudes (Tg NO yr$^{-1}$) of NO$_x$ emissions in the CEDS (black) and MACCity (red) anthropogenic emissions inventories.**

5. Section 3.3. In Figure 5 we see a site in eastern China with negative surface ozone trend that is not consistent with other site in the East Asia. Is this negative trend spread across all seasons? Is there any previous report that explains the negative trend?

This site is LinAn in China. This is a small, insignificant negative trend (-0.3 ppb decade$^{-1}$). This negative trend is primarily due to a large drop in ozone spanning from 2006-2010. These decreases occur most strongly in fall, when air masses reaching the site are dominated by air masses from the north and marine areas (Xu et al., 2008). The decreases are likely due to transport of clean marine air that does not reflect the growing urban emissions in China. Wu et al. (2008) also suggest a decreasing trend at the surface in LinAn. We add a sentence to discuss this in Section 3.3:

"Including the United States and Europe sites, we find that 42% of global surface background sites (114 of 271) show ozone increases since the 1990s, with notable decreases at 48 of the 52 United States sites and 100 of 186 Europe sites (Fig. 5) due to emissions regulations. Decreases

in eastern China (LinAn) can be attributed to the prevalence of clean marine air masses impacting that site during fall that do not reflect the growing urban emissions in China (Xu et al., 2008)."

6. Section 3.4, Line 489-492: I didn't get clear information from the analyses at WLG site. The authors argue that "expect similar trends in the Japan FT since increasing ozone over Japan is influenced by transport from China". But WLG is located at the west of the eastern China (not influenced by polluted outflow there), while the Japanese sites may by affected by the outflow but should mostly be constrained in springtime, why we should expect similar ozone trends at both sites?

The Reviewer is correct that we have overstated this and should not compare the Mt. Waliguan site to our Japanese ozonesondes, as they are likely to be impacted by different air masses. We have removed the discussion of the Japanese ozonesonde sites in relation to Mt. Waliguan and leave Mt. Waliguan as a standalone analysis. The paragraph in Section 3.4 now reads:

"Two mountaintop sites influenced by FT air are Mauna Loa (Hawaii) and Mt. Waliguan (China). At both of these sites, FT trends measured at the mountaintop sites show increasing FT (700-400 hPa) ozone trends. At Mt. Waliguan, FT trends can be isolated using nighttime ozone values, and measurements show an increase in FT ozone of $2.8 \pm 1.6$ ppb decade$^{-1}$ from 1994-2013 (Xu et al., 2016) and $1.7 \pm 0.5$ ppb decade$^{-1}$ from 1994-2016 (Cooper et al., 2020). This finding is attributed to both transport of increasing anthropogenic emissions and intensifying STE, which can explain 60% of the springtime ozone increase (Xu et al., 2016, 2018). While we do not analyze any ozonesonde launch locations over China and therefore do not have a direct comparison to sonde information, it is important to recognize the pattern of increasing FT ozone at multiple sites throughout the globe."

7. Line 497: where does "up to 1.7 ppbv per decade" come from?

This refers to the upper end of the range of trends we found at Hilo. The trends we found across the pressure levels evaluated ranged from -0.2 to 1.7 ppb decade$^{-1}$. We have rephrased this to clarify: "The trend reported in Cooper et al. (2020), which best matches our analysis timeframe, is higher than the average FT trends we calculated over Hilo from ozonesonde measurements from 1990-2017 ($0.9 \pm 0.6$ ppb decade$^{-1}$ from 700-400 hPa), but falls within the range measured in the FT (range of -0.2 to 1.7 ppb decade$^{-1}$)."

8. Line 517: What does "primary influences" mean?

We are referring to the main driver of the ozone trend at each location on Antarctica. We have rephrased this for clarification in Line 592-594: "Differences in the increases at these two stations may occur as a function of station location and whether anthropogenic sources or meteorological variables are the main drivers of ozone trends at each station."

9. Section 3.5. I few that the title of "Drivers of observed ozone change" is misleading. I am not sure whether the change of 5$^{th}$ ozone alone can be used to indicate ozone drivers of emission or transport. Why not integrate this part with the model sensitivity studies? In particular, I am not

convinced by the statement in Line 549 that "multiple previous analyses suggest that regional ozone increases are best explained by transport". Transport from where? Why transport is a more likely reason than NOx reduction to explain 5th ozone increase in US and Europe?

We have changed the title of the section to "Potential drivers in 5th percentile ozone" to avoid overstating our results. We include this section among our observational results to be clear that we are not using our own models to understand the increases in 5th-percentile ozone. We only discuss our model results in terms of their ability to reproduce observations – that is, we are not trying to diagnose trend drivers with our model evaluation. Rather, we use the sensitivity studies as a way of exploring reasons why models struggle to reproduce ozone trends.

The purpose of this section was to discuss the multitude of potential drivers behind the observed increasing ozone trends, and we emphasize that part of the increase we see in our ozonesonde results is due to an increase in low-percentile ozone. Here, we are pointing out that this increase in low-percentile ozone is consistent with other observational and modeling analyses that point to transport of ozone and its precursors from the tropics to the mid-latitudes as a major driver of these increasing trends at non-urban locations. NOx reductions may also contribute to increasing 5th-percentile ozone, but this is more specific to urban locations, which we avoid in this analysis. We now clarify that the impact of NOx reductions is mostly seen locally in urban areas, not the baseline sites examined in this analysis. For rural locations not impacted by local emissions, it is less likely that decreased NOx emissions in urban areas explains the increase in low-percentile ozone. Our changes in Section 3.5:

"Increasing 5th percentile concentrations are consistent with other analyses that suggest baseline ozone has been increasing, especially in the Northern Hemisphere. Increases in 5th percentile ozone have been attributed to a number of factors: decreased titration from $NO_x$ as a result of emissions decreases on a local scale, especially over urban areas in the United States and Europe (Yan et al., 2018b; Simon et al., 2015; Lin et al., 2017; Gao et al., 2013; Clifton et al., 2014), increases in methane concentrations (Lin et al., 2017), changes to large-scale processes such as STE (Parrish et al., 2012), and transport of ozone from the tropics and subtropics (Zhang et al., 2016; Gaudel et al., 2020). While all of these factors likely play a role in increased 5th percentile ozone in the Northern Hemisphere, multiple previous analyses suggest that regional, baseline ozone increases observed in rural locations with little impact from local emissions are best explained by transport from the tropics (Zhang et al., 2016, 2021). The largest emissions of ozone precursors have shifted toward low-latitude nations, especially in Southeast, East, and South Asia, where increased convection and temperature lead to more efficient ozone production compared to the mid-latitudes. This ozone is then transported poleward (Zhang et al., 2016). Tropospheric ozone increases in the middle troposphere (550 to 350 hPa) over mid-latitudes can be largely explained in models through transport of ozone from low latitudes, with STE playing an important role in the upper troposphere (above 350 hPa) (Zhang et al., 2016; Gaudel et al., 2020). Only 15% of the ozone increase over the western US between 1980-2014 has been attributed to an increase in methane concentrations (Lin et al., 2017)."

10. Line 566-567: Again, it would be important to compare the emission trends used in GMI and GC.

As discussed in Point 4, we do not think that differences in the emissions inventories are the main reason for differences in ozone trends between models. MACCity generally shows smaller trends in NOx than CEDS, but the MERRA2-GMI simulation captures larger ozone trends and concentrations than GEOS-Chem. The general pattern between the two inventories is the same: increasing global trends in NOx emissions.

11. Section 5.1. Line 690-691. Wang et al. (2022, in review) shows that aircraft emissions may be an important source of trend underestimation in chemical models.

We thank the Reviewer for bringing this to our attention. We have included in Section 5.1 a brief discussion of the Wang et al. paper and their finding that aircraft emissions may be another source of trend underestimation in models:

"Previous analyses have identified significant challenges facing models in reproducing observed tropospheric ozone trends in recent decades (Parrish et al., 2014; Young et al., 2018). Another analysis using GEOS-Chem from 1995-2017 found that the model underestimated global ozone trends compared to aircraft measurements and that aircraft emissions are a potential source of trend underestimates in the model (Wang et al., 2022)."

12. Section 5.2. I am curious about the stratospheric influence on tropospheric ozone. First, are both models show consistent pattern of ozone trends in the upper troposphere and lower stratosphere? A latitude-altitude plot of modelled ozone trends from the two models would be plausible. Second, I wonder what processes may contribute to increasing STE in the GMI model, is it more likely due to recover of ozone hole or changes in STE dynamics?

First, we find that both models show consistent patterns in ozone trends in the stratosphere. Below is a plot that shows the trends predicted by all models at 120, 100, and 50 hPa. Both GEOS-Chem and MERRA2-GMI show similar trends. The 2x2.5-degree GC resolution matches MERRA2-GMI more closely due to its finer resolution. Thus, it is likely that model representation of stratosphere-troposphere exchange plays an important role in model free tropospheric ozone trends, as resolution highly impacts vertical transport.

[Figure]

**Figure S10. Stratospheric trends (ppb decade-1) in GEOS-Chem and MERRA2-GMI. Trends are plotted as a function of ozonesonde launch latitude.**

Your second question is excellent and also difficult to answer. A previous study suggests that, from 1994-2010, STE has increased and driven a small increase in tropospheric ozone burden (Griffiths et al., 2020). This increase is attributed to stratospheric ozone recovery. MERRA2-GMI captures a decrease in lower stratospheric ozone in the northern extratropics (20-60° N) from 1998-2016, when ozone-depleting substances are no longer increasing (Wargan et al., 2017), and this decrease is offset by an increase in upper stratospheric ozone due to ozone recovery. The decreasing trend in the LS is attributed to changes in lower stratospheric ozone circulation that intensifies transport of ozone-poor air from the tropics to the extratropics. Circulation changes may be due to climate change, but this is unclear. It appears both changing STE dynamics and recovery of the ozone hole impact stratospheric trends in MERRA2-GMI,

and the relative impact of both is currently difficult to quantify. We add text to Section 5.2 to discuss these items and include the above figure in the SI (Figure S10):

"Figure 12, which investigates the role of STO3 in explaining ozone trends in MERRA2-GMI, also shows the importance of transport for understanding ozone trends. At 400 hPa over Europe, North America, and the NH Polar region, ozone trends are largely attributable to the stratospheric ozone influence. This aligns with the GEOS-Chem sensitivities that suggest meteorological inputs drive model trends at 400 hPa. Stratospheric influence is also prevalent at lower pressure levels for Europe and North America, consistent with previous analyses of ozone trends over these regions (Liu et al., 2020; Ordóñez et al., 2007). At the surface, the influence of STE is negligent in all regions. Importantly, MERRA2-GMI captures trends at 400 hPa remarkably well in Europe, North America, and the NH Polar region, which can be attributed to the ability of MERRA2-GMI to capture STE, likely due to its high resolution (Knowland et al., 2017). Model ability to capture vertical transport is important in reproducing ozone trends. GEOS-Chem and MERRA2-GMI show similar stratospheric trends (Fig. S10) but different trends at 400 hPa (Figs. 11 & 12), which suggests that transport from the stratosphere is most important for capturing trends at 400 hPa. Increases in stratospheric ozone influence in the troposphere may stem from both changes in STE dynamics and recovery of the ozone hole. MERRA2-GMI has been shown to capture a decrease in lower stratospheric ozone in the northern extratropics from 1998-2016, when ozone-depleting substances were no longer increasing (Wargan et al., 2017). This decreasing trend was attributed to changes in lower stratospheric ozone circulation that may be due to climate change, but evidence for this is unclear. This decrease is offset by an increase in upper stratospheric ozone due to ozone layer recovery. The extent to which either dynamics or ozone recovery impacts increasing STO3 in MERRA2-GMI is currently difficult to quantify."

Reference:

Chang, K.-L., Schultz, M. G., Lan, X., McClure-Begley, A., Petropavlovskikh, I., Xu, X., and Ziemke, J. R.: Trend detection of atmospheric time series: Incorporating appropriate uncertainty estimates and handling extreme events, Elem. Sci. Anth., 9, 10.1525/elementa.2021.00035, 2021.

Chang, K.-L., Cooper, O. R., Gaudel, A., Allaart, M., Ancellet, G., Clark, H., Godin-Beekmann, S., Leblanc, T., Van Malderen, R., Nédélec, P., Petropavlovskikh, I., Steinbrecht, W., Stübi, R., Tarasick, D. W., and Torres, C.: Impact of the COVID-19 Economic Downturn on Tropospheric Ozone Trends: An Uncertainty Weighted Data Synthesis for Quantifying Regional Anomalies Above Western North America and Europe, AGU Adv., 3, e2021AV000542, 10.1029/2021AV000542, 2022.

Chang, K. L., Cooper, O. R., Gaudel, A., Petropavlovskikh, I., and Thouret, V.: Statistical regularization for trend detection: an integrated approach for detecting long-term trends from sparse tropospheric ozone profiles, Atmos. Chem. Phys., 20, 9915-9938, 10.5194/acp-20-9915-2020, 2020.

Gaudel, A., Cooper, O. R., Chang, K.-L., Bourgeois, I., Ziemke, J. R., Strode, S. A., Oman, L. D., Sellitto, P., Nédélec, P., Blot, R., Thouret, V., and Granier, C.: Aircraft observations since the 1990s reveal increases of tropospheric ozone at multiple locations across the Northern Hemisphere, Sci. Adv., 6, eaba8272, 10.1126/sciadv.aba8272, 2020.

Wang, H., Lu, X., Jacob, D. J., Cooper, O. R., Chang, K.-L., Li, K., Gao, M., Liu, Y., Sheng, B., Wu, K., Wu, T., Zhang, J., Sauvage, B., Nédélec, P., Blot, R., and Fan, S.: Global tropospheric ozone trends, attributions, and radiative impacts in 1995–2017: an integrated analysis using aircraft (IAGOS) observations, ozonesonde, and multi-decadal chemical model simulations, Atmos. Chem. Phys. Discuss. [preprint], https://doi.org/10.5194/acp-2022-381, in review, 2022.

This is an excellent paper, and certainly appropriate for publication in ACP. I have just a few points that the authors should address before publication.

My main concern is that the authors are a bit too casual about possible bias changes in the ozonesonde data and the way that may increase uncertainty in their derived trends. For example, the Japanese stations changed from the KI sonde to the ECC sonde in the 2000s. The authors cite Tanimoto et al. (2015) to claim "…but this switch did not impact long-term trends". That is unlikely; even a difference of a few percent in average response can induce a large error in calculated trends. Figure 5 of Tanimoto et al. does in fact indicate a difference of this magnitude. Minor instrument changes or changes to preparation procedures can also make significant differences, as the effect of data revisions shows (see, e.g. Figure 15 of Tarasick et al., 2016). These additional uncertainties are more difficult to estimate than the statistical uncertainty output by standard packages, but they should be considered and discussed.

The Reviewer is correct that we need further discussion of these uncertainties. ECC sensors, which are the most widely used sensors, exhibit an uncertainty of 3-5%, while CI sensors exhibit uncertainties from 5-10%. We also note that we now use homogenized data where possible in response to this comment and comments from the community (Table 1), and we ensure that the data that is not homogenized does not contain step changes (Figures S1-S2).

We note that 8 of our 25 sites switch sensors at some point during the time frame. Of these sites, 2 have already been homogenized by HEGIFTOM, reducing the errors associated with sensor switch. We expect that this switch will add extra uncertainty, which is difficult to estimate. However, we do not expect these sensor switches to impact our overall conclusions, as trends measured at other sites in those regions support the conclusion that ozone has largely been increasing over time. Further, we have analyzed all of our sites for step changes and do not note any in our data. We acknowledge that changing sensors in the middle of the timeframe could substantially impact trend magnitudes, and we add a discussion of this to the methods section (Section 2.1):

"Most ozonesonde data were measured by electrochemical concentration cell (ECC) sensors, widely regarded as the most accurate sensor type (Tarasick et al., 2021). Four sites (Payerne, Uccle, Legionowo, Lindenberg) in Europe switched from using Brewer-Mast (BM) sensors to ECC sensors partway through their data records, and data from both sensors were used since previous analyses showed good agreement between measurements (De Backer et al., 1998; Stübi et al., 2008). Only Hohenpeissenberg (Europe) used the BM sensor throughout the time period. Naha (Japan), Tsukuba (Japan), Sapporo (Japan), and Syowa (Antarctica) both used carbon iodine (CI) sensors prior to 2010 and ECC sensors after, and this switch could impact overall long-term trends (Tanimoto et al., 2015). Typical uncertainties for CI sensors range from 5-10%, while they are 3-5% for ECC sensors (Tanimoto et al., 2015). This could lead to substantial differences in calculated trends, and we discuss trends from these sites in the context of regional trends using sites with more reliable data (e.g., only one sensor type or homogenized data). We note that trends at these sites should be treated with caution. A full list of these sites is in Table 1."

Minor points:

Line 16: "…suggesting the importance of emissions in observed changes." Or of surface (land use) changes?

We have modified this sentence to read "…suggesting the importance of surface emissions (anthropogenic, soil $NO_x$, impacts on biogenic VOCs from land use changes, etc.) in observed changes."

Line 20: "…reflect the global increase of background ozone." Really? I would have thought this reflects the reduction of NOx due to emission controls. But this should then be more evident at sonde sites with more urban influence, so the authors could perhaps say something about this.

The Reviewer is correct that decreased titration from NOx would be primarily an urban impact. We now clarify that we are referring to rural areas, where the increase in 5[th]-percentile ozone is less likely to be attributed to urban NOx controls:

"In all regions, increasing ozone trends both at the surface and aloft are at least partially attributable to increases in 5[th] percentile ozone, which average $1.8 \pm 1.3$ ppb decade$^{-1}$ and reflect the global increase of baseline ozone in rural areas."

Lines 69-74: These are large differences. Can the authors offer any explanation for the wide range of estimates? Also, please quote trends either in per year or per decade: the mixture of units is confusing to the reader.

The large range in these values is likely due to the slightly different geographical areas examined in each study – for example, the Ding et al. (2008) study focused only on Beijing, while Gaudel et al. (2020) looked at Southeast Asia as a region. Further, each study focuses on a different date range, which also contributes to the spread in the trends. We have now included a brief statement that differences in trends are likely due to differences in geographical area and dates analyzed in each study. We have also changed our numbers to be reported solely as ppb decade$^{-1}$. This now reads as:

"Free tropospheric (FT) ozone changes are highly regional and are impacted by emissions and transport. Aircraft measurements from 1995-2015 suggest FT ozone has increased strongly over Southeast Asia (5.6 ppb decade$^{-1}$; 14% decade$^{-1}$) (Gaudel et al., 2020), which is largely attributed to emissions increases. Ziemke et al. (2019) found that ozone increased over East Asia by 1 DU decade$^{-1}$ from 1979-2005 (~25% decade$^{-1}$) via satellite measurements, consistent with Ding et al. (2008), who found that ozone increased over Beijing by 20% decade$^{-1}$ from 1995-2005 using aircraft measurements. Increases over Asia have occurred most rapidly starting in the mid-2000s (~6% yr$^{-1}$ ~60% decade$^{-1}$) (Oetjen et al., 2016; Ziemke et al., 2019). Differences in trends between these studies can be attributed to differences in geographical areas (e.g., Beijing vs. Southeast Asia) as well as date ranges."

Lines 79-81: "The impact of STE on tropospheric ozone trends is potentially substantial: the observed interannual variability of the Brewer-Dobson circulation in the stratosphere leads to

changes of ozone levels in the northern mid-latitudes of ~2% (Neu et al., 2014)." This sentence seems at first to contradict itself, especially since the point of the Neu et al. paper was that the impact of STE on tropospheric ozone variability was NOT substantial. The authors should make it clear that they are referring to short-term trends.

We agree that this is not the reference to use. The point we are trying to make here is that STE can impact tropospheric ozone trends. To that end, we now cite studies that have addressed the impact of stratospheric intrusion on decadal ozone trends. This impact is typically regional and can be small, but in other cases it is substantial.

"STE has also been shown in both observations and models to have a substantial impact on tropospheric ozone trends and interannual variability, with stratospheric intrusion events influencing decadal trends across North America, Europe, the Southern Pacific, and the southern Indian Ocean (Williams et al., 2019; Liu et al., 2020). For example, models suggest 25-30% of increases in surface ozone between 1980 and 2010 were attributable to STE in multiple regions (Williams et al., 2019) and >10% of interannual variability in surface ozone could be explained by stratospheric ozone in winter and spring in North America (Liu et al., 2020)."

Lines 121-122: The authors should also note the endpoints for the Tarasick et al., 2016 trends (1966/1980-2013). This may be one reason for "mixed" results.

Thank you for pointing this out. We now include the endpoints for the Tarasick trends and a sentence explaining that this may be the reason that the two Arctic site studies found different trends:

"Trends from ozonesondes over Canada show mixed results, where one analysis found ozone increases from 2005-2014 (Christiansen et al., 2017), and another found no significant trend from 1966-2013 (Tarasick et al., 2016). These differences are partially attributable to a difference in analysis timeframes."

Lines 135-138: "Parrish et al. (2014) and Staehelin et al. (2017) showed that four state-of-the-science chemistry-climate models overestimate the absolute ozone mixing ratio by 5-17 ppb at mid-latitude background sites and capture only about half of the observed ozone increase over the last five decades…". Actually, no. A much more comprehensive analysis by Tarasick, Galbally, et al. (2019), which examined biases in historical measurements in great depth, has found smaller increases in surface ozone, of the order of 50%, which are in general agreement with model predictions. In addition, the analysis of ice-core data by Yeung et al. (2019), which the authors cite, and the independent analysis of aircraft and balloon data by Tarasick, Galbally et al. (2019), also both support a smaller increase of surface ozone, of the order of 50%.

Tarasick et al. (2019) and Yeung et al. (2019) are both referring to long-term changes since the historical ozone record (i.e., 1900 or before). In Tarasick et al., this referred to changes since 1877, and Yeung et al. is referring to changes since 1900. Both of these timeframes are much longer than the one referred to in Parrish et al. (2014) and Staehelin et al. (2017), which are

focused on changes since ~1950. Parrish et al. and Staehelin et al. both find that models underestimate modern long-term ozone changes since 1950 by ~50%, and this does not necessarily contrast with findings from studies examining historical periods. Measurements used in Parrish et al. and Staehelin et al. should be more accurate than those used in the historical time range. The corrections made in Tarasick et al. focus on pre-1960s measurements and do not apply to the more accurate modern observations. For these reasons, we keep our discussion of Parrish et al. and Staehelin et al. as is, as their findings for the modern period are not contradicted by analyses of the historical period in Tarasick et al. and Yeung et al.

Line 186: How interpolated? Are these integrals between midpoints, or a point estimate?

"Interpolate" is perhaps not quite the right word to use. Here, rather than integrate, we aggregate ozone between pressure levels defined by GEOS-Chem. We clarify this in Line 230:

"Ozonesonde profiles were reduced to match the 47-layer GEOS-Chem reduced pressure levels by aggregating all observed ozone values between model-defined pressure edges."

Line 190: Figure 1 is nice, but a table identifying the ozonesonde sites, and the dates of their records (as well as breaks, for sonde type, or other reasons) would be very helpful.

We have included a table (Table 1) with this information, shown below.

**Table S1. Summary of all ozonesonde launch locations, dates, sensor types, data source, and region. Also included is whether each site has been homogenized.**

| Sonde Launch Location | Dates | Sensor Type | Homogenized? | Data Source | Region |
|---|---|---|---|---|---|
| Alert | 1990-2016 | ECC | Y | HEGIFTOM | NH Polar |
| Boulder | 1980-2016 | ECC | Y | NOAA | North America |
| Broadmeadows | 1999-2016 | ECC | N | WOUDC | Southern Hemisphere |
| De Bilt | 1993-2015 | ECC | Y | HEGIFTOM | Europe |
| Edmonton | 1980-2016 | ECC | Y | HEGIFTOM | North America |
| Eureka | 1993-2016 | ECC | Y | HEGIFTOM | NH Polar |
| Goose Bay | 1980-2016 | ECC | Y | HEGIFTOM | North America |
| Hilo | 1985-2015 | ECC | Y | SHADOZ | Hawaii |
| Hohenpeissenberg | 1980-2017 | BM | Y | HEGIFTOM | Europe |
| Lauder | 1986-2016 | ECC | Y | HEGIFTOM | Southern Hemisphere |
| Legionowo | 1980-2015 | BM, ECC since 1993 | N | WOUDC | Europe |
| Lerwick | 1994-2016 | ECC | N | WOUDC | Europe |

| | | | | | |
|---|---|---|---|---|---|
| Lindenberg | 1980-2013 | BM, ECC since 1992 | N | WOUDC | Europe |
| Macquarie Island | 1994-2017 | ECC | N | WOUDC | Southern Hemisphere |
| Naha | 1991-2016 | CI, ECC since 2008 | N | WOUDC | Japan |
| Nairobi | 1997-2016 | ECC | Y | SHADOZ | Southern Hemisphere |
| Neumayer | 1992-2014 | ECC | N | WOUDC | Southern Hemisphere |
| Ny Aalesund | 1990-2012 | ECC | N | WOUDC | NH Polar |
| Payerne | 1980-2016 | BM, ECC after 2002 | Y* | HEGIFTOM | Europe |
| Sapporo | 1993-2016 | CI, ECC since 2009 | N | WOUDC | Japan |
| Sodankyla | 1989-2006 | ECC | N | WOUDC | NH Polar |
| Syowa | 1982-2017 | CI, ECC since 2010 | N | WOUDC | Southern Hemisphere |
| Tateno | 1980-2016 | CI, ECC since 2009 | N | WOUDC | Japan |
| Uccle | 1980-2015 | BM, ECC since 1997 | Y | HEGIFTOM | Europe |
| Wallops Island | 1980-2016 | ECC | Y | HEGIFTOM | North America |

*Note that Payerne has been homogenized only since 2002, a timeframe too short for this analysis, so we use the original data that spans the full timeframe.

Figure 3 caption: It would be helpful to restate here that the points correspond to the 47-layer GEOS-Chem reduced pressure levels.

The caption now reads: "Trends (ppb decade$^{-1}$) through the free troposphere (800-400 hPa, reduced to GEOS-Chem pressure levels) at the 25 global ozonesonde sites with data from 1990-2017, distributed into six regions. Solid circles indicate that the trends are statistically significant ($p<0.1$), while open circles denote statistically insignificant trends."

Figure 4: How are the 800-400 hPa values calculated?

Here, we are using all ozone values from our reduced pressure levels spanning from 800 to 400 hPa, thus this is a representation of ozone shifts across the free troposphere. Although this is stated in the preceding paragraph, we reiterate in the figure caption that these density plots include all values in the pressure range 800-400 hPa, and the medians are calculated from this pool of values:

Figure 4 caption: "Changes in ozone concentration (ppb) distributions between the first five years of analysis (red; 1990-1994) and the last five years of analysis (blue; 2010-2014 for surface; 2013-2017 for sondes) shown as density functions at the surface (background sites compiled by TOAR) and throughout the troposphere (all ozone values measured by ozonesondes in the pressure range 800 to 400 hPa). Median concentrations are shown with vertical lines, and the corresponding values and number of sites are recorded inset."

Lines 453-454: "Our results are consistent with other global analyses of surface ozone data …" Some citations are in order here.

We have added extra citations to this sentence:

"Our results are consistent with other global analyses of surface ozone data that have shown increases over varying timeframes beginning in the 1990s at far fewer sites spanning a narrower slice of the globe (Cooper et al., 2020, 2014)."

Line 473: "baseline ozone". What is "baseline" ozone? The authors also refer to "background" ozone. Are they the same?

We have clarified here and in several other places what we mean by baseline ozone, which is that we are referring to ozone that is not influenced by local emissions. We have removed the term "background" except where it applies to policy-relevant background ozone as defined by the EPA.

Line 515: Note that Syowa also changed from CI to ECC sondes.

We have added this to our discussion of trends at Syowa:

"At the South Pole, increases are associated with ozone-rich air from the upper troposphere and lower stratosphere, whereas Syowa, located at 69° S, is primarily impacted by marine air and air-mass transport from regions near South America (Kumar et al., 2021). It is also important to note that Syowa switched sensors from CI to ECC in 2010, which could impact trends."

Lines 544-545: I'm not sure that the increases in 5th percentile ozone due to decreased titration from NOx would necessarily translate to global changes in ozone. This could be a localized effect, more evident at sonde sites with more urban influence.

The Reviewer is correct that an increase in baseline ozone due to a reduction of NOx would be primarily evident in urban areas, which we avoid in this analysis in favor of rural areas away from the influence of local sources. We now clarify in Section 3.5 that the impact of NOx reductions is mostly seen locally in urban areas, not the rural baseline sites examined in this analysis:

"Increasing 5$^{th}$ percentile concentrations are consistent with other analyses that suggest baseline ozone has been increasing, especially in the Northern Hemisphere. Increases in 5$^{th}$ percentile ozone have been attributed to a number of factors: decreased titration from $NO_x$ as a result of

emissions decreases on a local scale, especially over urban areas in the United States and Europe (Yan et al., 2018b; Simon et al., 2015; Lin et al., 2017; Gao et al., 2013; Clifton et al., 2014), increases in methane concentrations (Lin et al., 2017), changes to large-scale processes such as STE (Parrish et al., 2012), and transport of ozone from the tropics and subtropics (Zhang et al., 2016; Gaudel et al., 2020). While all of these factors likely play a role in increased 5$^{th}$ percentile ozone in the Northern Hemisphere, multiple previous analyses suggest that regional, baseline ozone increases observed in rural locations with little impact from local emissions are best explained by transport from the tropics (Zhang et al., 2021; 2016). The largest emissions of ozone precursors have shifted toward low-latitude nations, especially in Southeast, East, and South Asia, where increased convection and temperature lead to more efficient ozone production compared to the mid-latitudes. This ozone is then transported poleward (Zhang et al., 2016). Tropospheric ozone increases in the middle troposphere (550 to 350 hPa) over mid-latitudes can be largely explained in models through transport of ozone from low latitudes, with STE playing an important role in the upper troposphere (above 350 hPa) (Zhang et al., 2016; Gaudel et al., 2020). Only 15% of the ozone increase over the western US between 1980-2014 has been attributed to an increase in methane concentrations (Lin et al., 2017)."

Lines 681-684: As noted previously, the large Parrish et al. (2014) estimate of the ozone change has not been supported by subsequent analysis (Tarasick, Galbally et al., 2019; Yeung et al., 2019), so a finding that this is "consistent with our current analysis" may not be the best advertisement for the authors' work, especially to lead this section. However, Tarasick, Galbally et al. also found that "…the uncertainty in the estimated increases … depends more on the modern region chosen for comparison than on the historical data. Data representativeness [of modern data] thus seems to be the more important source of uncertainty." The authors may want to mention data representativeness; it's an important topic, and especially pertinent when one is using only a small number of observation sites for evaluation of models.

As discussed above, we keep the Parrish et al. and Staehelin et al. discussion in Section 5.1 since their findings relate to the modern period of observations, rather than the historical ones discussed in Tarasick et al. and Yeung et al.

Your point about the representativeness of ozone measurements is a very good one. We added a statement from Tarasick et al. (2019) in Section 5.1 about how representation of ozone in measurements around the globe adds limitations to understanding ozone's radiative forcing and trends over time. We also point out a recent study that shows model underestimation of global ozone trends. It now reads:

"Previous analyses have identified significant challenges facing models in reproducing observed tropospheric ozone trends in recent decades (Parrish et al., 2014; Young et al., 2018). In the northern hemisphere mid-latitudes, chemistry-climate models were only able to reproduce ~50% of the observed ozone trend (Parrish et al., 2014), consistent with our current analysis using chemical transport models. Another analysis using GEOS-Chem from 1995-2017 found that the model underestimated global ozone trends compared to aircraft measurements and that aircraft emissions are a potential source of trend underestimation in the model (Wang et al., 2022).

Tarasick et al. (2019) also pointed out the role of data representativeness: uncertainty in estimated observational trends stems largely from data representativeness rather than the accuracy of historical data, pointing to the importance of increasing ozone monitoring station number and frequency, especially when the evaluation of model skill necessarily relies on comparison to sparse datasets."

Line 688: Characterizes? Something is wrong with this sentence.

We have changed this sentence to now read: "Our findings that models are not able to reproduce recent ozone trends contrast with an analysis of GEOS-Chem and GISS-E2.1 that found the model accurately reproduced preindustrial ozone concentrations (Yeung et al., 2019)."

**Citation**: https://doi.org/10.5194/acp-2022-330-RC2

---

## Referee Report (RR1)

Review of "Multidecadal increases in global tropospheric ozone derived from ozonesonde and surface site observations: Can models reproduce ozone trends?" by A. Christiansen, et al.

-Review by Ryan Stauffer, NASA/GSFC, as a follow-up to public comment https://doi.org/10.5194/acp-2022-330-CC1

General Comments:

I sincerely thank the authors for addressing all of our public comments, as well as the two formal reviews provided to ACPD. The authors invested significant effort to ensure that the most up to date and accurate versions of available (i.e., homogenized) ozonesonde data were used. For example, the Canadian station trends in Figure S4 show changes that I expected based on experience working with both non-homogenized and homogenized versions of those data. The additional discussion, analyses, and arguments certainly satisfy all of my comments and concerns. I only have a few remaining minor/technical comments, and I think this paper is well-suited for publication in ACP.

Minor Comments:

Table 1: Wallops Island ozonesonde data have only been homogenized back to 1995 (https://agupubs.onlinelibrary.wiley.com/doi/full/10.1029/2018JD030098), so please double check these date ranges and/or specify when stations have not completely homogenized their records.

Line 204 and Figure S1: The Japanese stations certainly appear to have a notable step-change and increase associated with the move from CI to ECC ozonesondes around 2010. Is there also a step-change in stratospheric ozone at these stations at the same time, indicating a large overall change in sensor response? If so, then indeed caution should be used in interpreting these large positive trends in the troposphere.

Line 351: "I other simulation…" looks like a typo

Line 847: "negligible" rather than "negligent"

A note on the Stauffer et al., (2020) study on the ozonesonde "dropoff": An updated analysis is now in press with Earth and Space Science. However, no additional stations are considered "affected" by the dropoff, so this is purely for your information:

Stauffer, R. M., Thompson, A. M., Kollonige, D. E., Tarasick, D. W., Van Malderen, R., Smit, H. G. J., et al. (2022). An Examination of the Recent Stability of Ozonesonde Global Network Data. Earth and Space Science, 9, e2022EA002459. https://doi.org/10.1029/2022EA002459

---

## Author Response (AR2)

Review of "Multidecadal increases in global tropospheric ozone derived from ozonesonde and surface site observations: Can models reproduce ozone trends?" by A. Christiansen, et al.
-Review by Ryan Stauffer, NASA/GSFC, as a follow-up to public comment https://doi.org/10.5194/acp-2022-330-CC1

General Comments:
I sincerely thank the authors for addressing all of our public comments, as well as the two formal reviews provided to ACPD. The authors invested significant effort to ensure that the most up to date and accurate versions of available (i.e., homogenized) ozonesonde data were used. For example, the Canadian station trends in Figure S4 show changes that I expected based on experience working with both non-homogenized and homogenized versions of those data. The additional discussion, analyses, and arguments certainly satisfy all of my comments and concerns. I only have a few remaining minor/technical comments, and I think this paper is well-suited for publication in ACP.

Minor Comments:
Table 1: Wallops Island ozonesonde data have only been homogenized back to 1995 (https://agupubs.onlinelibrary.wiley.com/doi/full/10.1029/2018JD030098), so please double check these date ranges and/or specify when stations have not completely homogenized their records.

Thanks for finding this typo. We have fixed the date ranges in Table 1 (reproduced below).

**Table 1. Summary of all ozonesonde launch locations, dates, sensor types, data source, and region. Also included is whether each site has been homogenized.**

| Sonde Launch Location | Dates | Sensor Type | Homogenized? | Data Source | Region |
|---|---|---|---|---|---|
| Alert | 1990-2016 | ECC | Y | HEGIFTOM | NH Polar |
| Boulder | 1980-2016 | ECC | Y | NOAA | North America |
| Broadmeadows | 1999-2016 | ECC | N | WOUDC | Southern Hemisphere |
| De Bilt | 1993-2015 | ECC | Y | HEGIFTOM | Europe |
| Edmonton | 1980-2016 | ECC | Y | HEGIFTOM | North America |
| Eureka | 1993-2016 | ECC | Y | HEGIFTOM | NH Polar |
| Goose Bay | 1980-2016 | ECC | Y | HEGIFTOM | North America |
| Hilo | 1985-2015 | ECC | Y | SHADOZ | Hawaii |
| Hohenpeissenberg | 1980-2017 | BM | Y | HEGIFTOM | Europe |
| Lauder | 1986-2016 | ECC | Y | HEGIFTOM | Southern Hemisphere |
| Legionowo | 1980-2015 | BM, ECC since 1993 | N | WOUDC | Europe |
| Lerwick | 1994-2016 | ECC | N | WOUDC | Europe |
| Lindenberg | 1980-2013 | BM, ECC since 1992 | N | WOUDC | Europe |

| | | | | | |
|---|---|---|---|---|---|
| Macquarie Island | 1994-2017 | ECC | N | WOUDC | Southern Hemisphere |
| Naha | 1991-2016 | CI, ECC since 2008 | N | WOUDC | Japan |
| Nairobi | 1998-2016 | ECC | Y | SHADOZ | Southern Hemisphere |
| Neumayer | 1992-2014 | ECC | N | WOUDC | Southern Hemisphere |
| Ny Aalesund | 1990-2012 | ECC | N | WOUDC | NH Polar |
| Payerne | 1980-2016 | BM, ECC after 2002 | Y* | HEGIFTOM | Europe |
| Sapporo | 1993-2016 | CI, ECC since 2009 | N | WOUDC | Japan |
| Sodankyla | 1989-2006 | ECC | N | WOUDC | NH Polar |
| Syowa | 1982-2017 | CI, ECC since 2010 | N | WOUDC | Southern Hemisphere |
| Tateno | 1980-2016 | CI, ECC since 2009 | N | WOUDC | Japan |
| Uccle | 1980-2015 | BM, ECC since 1997 | Y | HEGIFTOM | Europe |
| Wallops Island | 1995-2016 | ECC | Y | HEGIFTOM | North America |

*Note that Payerne has been homogenized only since 2002, a timeframe too short for this analysis, so we use the original data that spans the full timeframe.

Line 204 and Figure S1: The Japanese stations certainly appear to have a notable step-change and increase associated with the move from CI to ECC ozonesondes around 2010. Is there also a step-change in stratospheric ozone at these stations at the same time, indicating a large overall change in sensor response? If so, then indeed caution should be used in interpreting these large positive trends in the troposphere.

Here, we remake Figure S1 to show annual median tropospheric ozone concentrations at the non-homogenized sites, with annual median stratospheric ozone shown for the Japanese sites. The stratospheric data for the Japanese sites are shown in the bottom 3 panels of Figure S1. In the stratosphere, the Japanese sites do not appear to have the type of step change around 2010 that is apparent in the troposphere. Here, we see that stratospheric ozone medians, while increasing, do not exhibit the drastic nature of the step changes in the troposphere. Since none of the Japanese stations have been homogenized, it is currently difficult to assess the extent to which a potential step change influences our results, but the fact that we see the step change occurring just in the troposphere implies that our derived ozone trend is not solely due to a large overall change in sensor response. We add a statement in our Results cautioning against over-interpreting the tropospheric trends:

"The strongest increasing trends from 1990-2017 occur in Japan, averaging 3.8 ± 0.8 ppb decade$^{-1}$ (7.1% ± 1.5% decade$^{-1}$) across all pressure levels and ranging from 2.4 to 5.3 ppb decade$^{-1}$ (4.4% to 9.9% decade$^{-1}$). Caution should be taken to not over-interpret the Japanese trends, as a potential step change occurs at these sites around 2010 in the troposphere (Fig. S1). While this may be partially due to a

change in sensor response, these step changes are not visible in the stratosphere (Fig. S1), suggesting that these trends mostly reflect the rapid increase in emissions over Asia in the past 4 decades."

[Figure]

**Figure S1. Annual median ozone profiles for ozonesonde data from the 12 non-homogenized sites from 1990-2017. No step changes are apparent in the data, with the exception of the Japanese sites (Naha, Sapporo, Tateno (Tsukuba)). These step changes are not apparent in the stratosphere.**

Line 351: "I other simulation..." looks like a typo

We have corrected this to read "The other simulation…"

Line 847: "negligible" rather than "negligent"

We have corrected this to be "negligible."

A note on the Stauffer et al., (2020) study on the ozonesonde "dropoff": An updated analysis is now in press with Earth and Space Science. However, no additional stations are considered "affected" by the dropoff, so this is purely for your information:
Stauffer, R. M., Thompson, A. M., Kollonige, D. E., Tarasick, D. W., Van Malderen, R., Smit, H. G. J., et al. (2022). An Examination of the Recent Stability of Ozonesonde Global Network Data. Earth and Space Science, 9, e2022EA002459. https://doi.org/10.1029/2022EA002459

Thank you for passing this along! We have added this reference to our Methods section, where we discuss the dropoff: "A recent study showed a drop in total column and stratospheric ozone measured by ECC instruments compared to satellite observations in the latter parts of their records for reasons still under investigation (Stauffer et al., 2020, 2022). We find that 5 of our 25 sites were impacted by these ozone measurement drops, although these drop-offs were typically limited to pressures above ~50 hPa, so our results should not be affected. Out of an abundance of caution, at these impacted sites, we used only data from before the unexplained sharp drop-off in ozone concentrations, as data before these drops is still considered highly reliable (Stauffer et al., 2020, 2022), and this resulted in the removal of up to one year of data at each affected site."